# *Operando*-electrified solvay process

Qi Huang[1], Jingjing Duan ©[2], Markus Antonietti ©[3] & Sheng Chen ©[1,3] ✉

Replacing energy- and cost-intensive Solvay process with innovative manufacture protocol, such as *operando*-electrified synthesis, would provide a simpler, scale-flexible pathway to produce $NaHCO_3$. However, the commercialization of such technique is hampered by the bottleneck problem of small productivity, which is two orders of magnitude below traditional Solvay process. This might be attributed to the gap in conditions between embedded processes inside the system, especially low concentrations of local alkaine, which are generated relatively slowly from one-electron hydrogen chemistry, leading to a small $CO_2$–to–$HCO_3^-$ conversion ratio. Guided by octet rule, here we embed eight-electron nitrogen chemistry into Solvay process, maximizing local alkalines generation for $CO_2$ conversion. By further breaking the stumbling scaling relationship with a liquid metal-derived catalyst, the system achieves productivity up to $3.63 \, mol \, L^{-1} \, h^{-1}$.

Conversion of carbon dioxide ($CO_2$) into valuable commodities is a hot topic. Other than extensively reported products like syngas and $C_{2+}$ hydrocarbons, another case in point is sodium bicarbonate ($NaHCO_3$), one of the 100 most important chemical compounds in the world with diverse applications[1], its global production being over 65 million metric tons (US$21.5 billion market value) annually[2]. Currently, over 60% of $NaHCO_3$ is produced by Solvay process[3], wherein carbon feedstock (like $CO_2$) is vented into saturated sodium chloride (NaCl) solution mixed with performed alkalines, leading to the precipitation of $NaHCO_3$ following solubility product principle:

$$NH_3 + CO_2 + H_2O \rightarrow NH_4HCO_3 \tag{1}$$

$$NH_4HCO_3 + NaCl \rightarrow NaHCO_3 \downarrow + NH_4Cl \tag{2}$$

$$2NH_4Cl + CaO \rightarrow 2NH_3 + CaCl_2 + H_2O (\text{Recycle}) \tag{3}$$

It is estimated that this production sector consumes ~4 million tons of $CO_2$ every year, but at the expense of huge energy consumption of $2.6 \sim 3.4 \times 10^{10}$ kWh. For each ton of $NaHCO_3$, ~10 $m^3$ of waste liquid solution has been produced and discharged into the sea. Chemicals in the waste ($Cl^-$ 850 - 1100 kg, $Na^+$ 160 - 220 kg, $Ca^{2+}$ 340 - 400 kg and suspended solids 90 - 700 kg) would cause serious environmental issues. Therefore, it is an urgent demand to transform into an innovative manufacturing protocol with high efficiency and low production cost (Fig. 1a)[4–6].

Among various components inside Solvay systems, alkaline is underpinned as a crucial feedstock for increasing $CO_2$ dissolution (only $1.45 \, g \, L^{-1}$ in aqueous solution), and according to Le Chatelier principle, shifting the reaction equilibrium toward $CO_2$ conversion. Nevertheless, all reported traditional/improved Solvay processes employ performed alkalines like $NH_3$ and hydroxyl ($OH^-$)[7–15], wherein the productivity is very sensitive to alkaline/$CO_2$ ratios. To achieve improved efficiencies, excessive alkalines are frequently used to increase $CO_2$ dissolution, leading to substantial environmental and economic costs. Moreover, these performed alkalines are produced/transported from other energy-intensive industries, such as $NH_3$ from the Haber-Bosch process[16] and hydroxyl ($OH^-$) from the chlor-alkali process[17], which has further complicated overall synthetic scheme.

By contrast, alkaline can also be generated as a side product in electrochemical reactions. By integrating these reactions into Solvay system, it can eliminate the use of performed alkalines, forming a synthetic scheme of *operando*-electrified process as follows:

$$CO_2 + operando \text{ alkalines} \rightarrow HCO_3^- \tag{4}$$

[1]Key Laboratory for Soft Chemistry and Functional Materials, School of Chemistry and Chemical Engineering, Nanjing University of Science and Technology, Ministry of Education, Nanjing 210094, China. [2]MIIT Key Lab Thermal Control Electronic Equipment, School of Energy and Power Engineering, Nanjing University of Science and Technology, Nanjing 210094, China. [3]Max Planck Institute of Colloids and Interfaces, Potsdam 214476, Germany. ✉e-mail: sheng.chen@njust.edu.cn

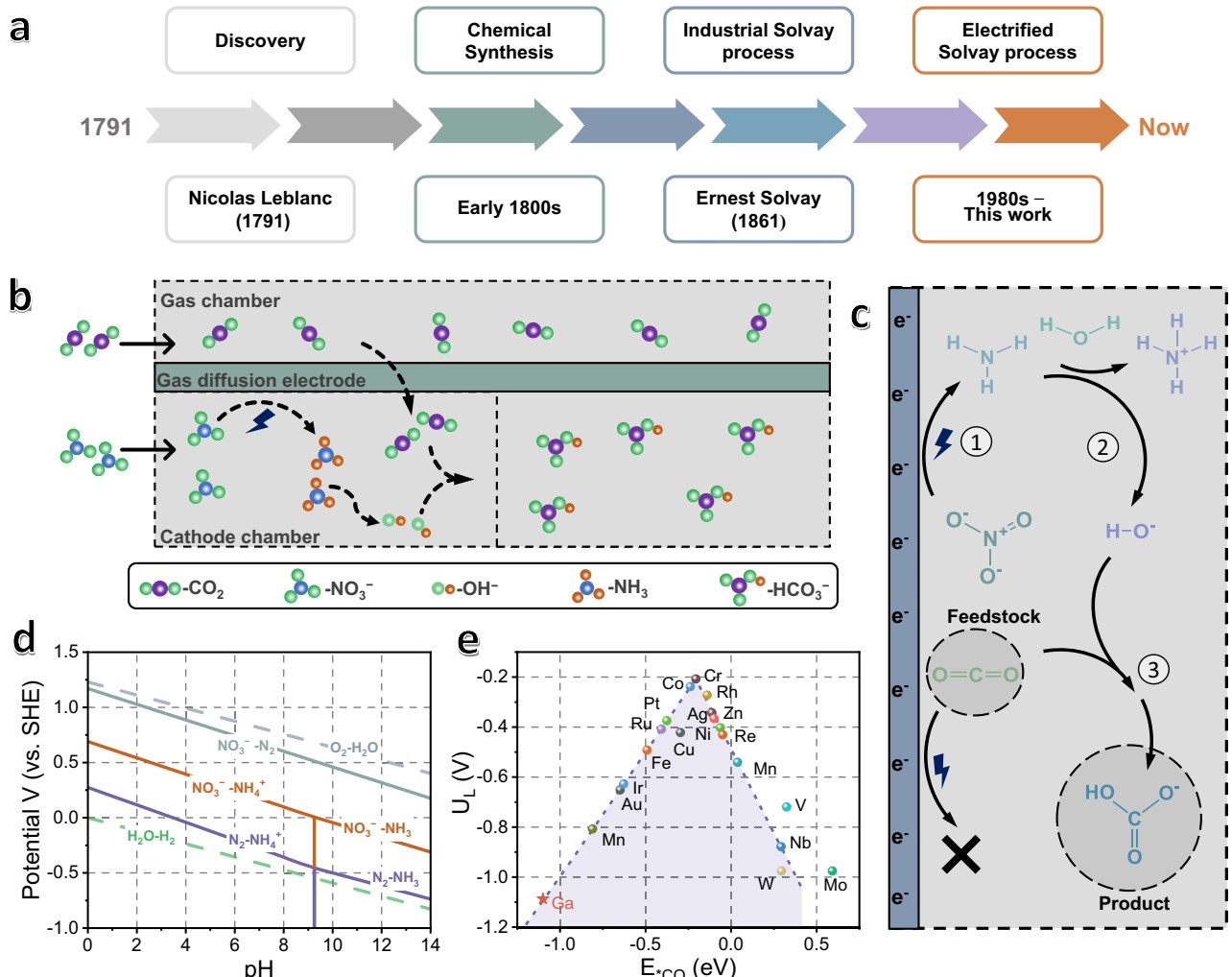

**Fig. 1 | Schematic diagram of HCO$_3^-$ production. a** The history of NaHCO$_3$ synthesis. **b** the electrofixation of CO$_2$ and NO$_3^-$ in flow cells. **c** The sketch mechanism for CO$_2$ and NO$_3^-$ electrofixation. **d** Pourbaix diagram for electrochemical NO$_3^-$ converting to NH$_3$ (Reproduced with standard electrode potentials in water at 298.15 K from ref. 28). **e** The volcano plot of *CO descriptor for common metal elements (Other than Ga, the other data is reproduced from ref. 29). Source data for (**d**, **e**) are provided as a Source Data file.

Such a synthetic scheme would be very attractive, as the only inputs required would be CO$_2$, *operando* alkalines (come from electrochemical reactions) and electricity. By exploiting the inherent flexibility of *operando* electrified systems, NaHCO$_3$ production could be carried out at either small or large scale in a one-step synthetic procedure with simplified configuration and reduced operation cost. However, in the preliminary examples of *operando* alkalines generate from hydrogen evolution reaction (HER): H$_2$O + e → 1/2H$_2$ + OH$^-$; E$^0$ = 0 V versus the reversible hydrogen electrode (vs. RHE), the NaHCO$_3$ productivity is prohibitively low, which is two orders of magnitude below benchmark Solvay process (0.79 mol L$^{-1}$ h$^{-1}$; Supplementary Table 1). This phenomenon might be attributable to the low concentrations of local alkaline, which is generated relatively slowly from one-electron hydrogen chemistry (HER)[14,18,19], and readily diffuses away with reaction flow, leading to a small CO$_2$ conversion ratio of H$_2$O/HCO$_3^-$ = 1/1 (H$_2$O + CO$_2$ + e → 1/2H$_2$ + HCO$_3^-$; E$^0$ = 0 V). Although large polarized potentials can accelerate the kinetics of *operando* HER, it would simultaneously promote parasitic CO$_2$RR (E$^0$ = −0.12 ~ 0.14 V) that consume the protons/electrons otherwise for producing alkaline. In line with this hypothesis, developing an innovative method to maximize alkaline generation, and thereby achieving a high local concentration of *operando* alkaline, should promote productivity in the *operando*-electrified

process. However, this strategy is proven to be challenging and rarely demonstrated.

In this work, we leverage "octet rule"[20], a fundamental principle in elemental chemistry, to address the above challenging problem. Generally, the reactivity of a chemical process is intricately linked to the number of electrons involved in migration. Other than hydrogen chemistry, such as HER, a vast range of elements in the periodic table adhere to the octet rule, allowing them to accommodate up to eight electrons in valence shells. This characteristic could facilitate maximally eight-electron chemical reactions. For instance, in nitrogen chemistry, the nitrate (NO$_3^-$) reduction reaction exemplifies this principle[21]. NO$_3^-$ reduction reaction (NO$_3$RR) affords eight-electron transport by generating the alkalines of NH$_3$ and OH$^-$ (NO$_3$RR: NO$_3^-$ + 6H$_2$O + 8e → NH$_3$ + 9OH$^-$; E$^0$ = 0.69 V)[22]. Consequently, by blending *operando* nitrogen chemistry (i.e., *operando* NO$_3$RR) into electrified process, it can theoretically achieve the maximum CO$_2$ conversion ratio of NO$_3^-$/CO$_2$ = 1/9 (NO$_3^-$ + 8e + 6H$_2$O + 9CO$_2$ → NH$_3$ + 9HCO$_3^-$). Motivated by the hypothesis, we have developed an *operando*-electrified system of delivering two environmental pollutants (CO$_2$ and NO$_3^-$) into an electrochemical cell to produce NaHCO$_3$ (Fig. 1b, c) despite its productivity still unable to rival Solvay process, owing to the stumbling scaling relationship (the relationship between productivity and production scale) inside the updated

system, i.e., parasitic processes of $CO_2RR$[23,24], HER ($E^0 = 0$ V)[25] and carbon−nitrogen (C−N) coupling reaction (CNR; Supplementary Table 2; $E^0 = 0.15 - 0.77$ V)[26,27].

Indeed, breaking scaling relationship is very difficult in the literature. In this study, we aim to modify traditional production methodologies and propose an *operando*-electrified production model to enhance the compatibility of Solvay process. Here, our density function theory (DFT) simulations further suggest a gallium (Ga)-derived catalyst, where its *CO binding energy is situated at a position far from the volcano apex, indicating it is inert to parasitic processes ($CO_2RR$ and CNR). In this *operando*-electrified system with a Ga-derived catalyst, we reveal an *operando*-positive-coupling phenomenon of reversing the negative impact of the scaling relationship into positive synergy, which expedites $CO_2$ dissolution, leading to high local alkaline concentration and superior $NaHCO_3$ productivity.

## Results

### Requirement for $CO_2$ and nitrate fixation

The first requirement for fixation is the favorable occurrence of $NO_3RR$ to produce $NH_3$ and $OH^-$, which facilitates $CO_2$ dissolution and conversion. As evidenced by Pourbaix diagram (Fig. 1d)[28], $NO_3RR$ can take place in the whole pH range from 0 - 14 with thermodynamic reaction potentials from 0.69 to −0.31 V (vs. SHE). When the pH of the system is above 9.25, the product is predominantly $NH_3$. This will increase the dissolution of $CO_2$ to provide more feedstock for the reaction. Interestingly, the inherently high thermodynamic potential of $NO_3RR$ can inhibit HER side process ($H_2O – H_2$ pairs, or HER), which allows for high reaction selectivity.

The second requirement for fixation is a suitable catalyst that prevents $CO_2$ from participating in the electrochemical reaction of $CO_2RR$ and CNR. Accordingly, a volcano diagram has been built to predict the activities of common metal elements using *CO adsorption energy as descriptors (Fig. 1e)[29]. According to Sabatier principle[30], on the right-hand side of the volcano plot are metal elements with weak adsorption ability for *CO intermediates, while on the left-hand side are those with strong adsorption, both of which lead to compromised $CO_2RR$ activities. It is those metal elements close to the summit of the volcano that display the best $CO_2RR$ activities. Here, we have calculated the *CO adsorption energy of liquid metal gallium (Ga), and found it situated at a distance far from the volcano apex ($E_{*CO} = -1.10$ eV and corresponding to the limiting potential of −1.09 V); therefore, it would be an inert candidate for $CO_2RR$. Nevertheless, as will be discussed later, Ga active sites have still displayed a physical adsorption to $CO_2$, which can serve to enrich $CO_2$ on the electrode surface and facilitate $CO_2$-to-$HCO_3^-$ conversion.

### Catalyst fabrication and electrochemical properties

A Ga-derived catalyst has been synthesized by dispersing bulk Ga in mixed solvents, followed by anchoring on rGO through hydrothermal treatment (i.e., GaOOH/rGO catalyst; please see the methods section). Morphology characterizations clearly demonstrate the uniform loading of GaOOH nanoparticles of 30 nm size on the surfaces of rGO nanosheets, which is different from individual GaOOH or rGO (Supplementary Figs. 1–3). The coexistence of each component inside GaOOH/rGO is verified by characterizations including high-resolution transmission electron microscopy (HR-TEM) showing the lattice fringes of 0.2384 nm for GaOOH (111) crystal (Fig. 2a; Supplementary Fig. 4) and X-ray diffraction (XRD) showing the characteristic diffraction peaks of both GaOOH and rGO (Fig. 2b; Supplementary Fig. 5). Inside the composite, the valence state of Ga has been determined as +3 by X-ray photoelectron spectra (XPS) (Fig. 2c) according to Ga $2p_{3/2}$ and $2p_{1/2}$ peaks at 1118.8 eV and 1145.8 eV, respectively. Some carboxyl groups also exist inside GaOOH/rGO, originated from rGO that serve as the anchoring sites for immobilizing GaOOH

nanoparticles (Fig. 2c; Supplementary Figs. 6–8)[31]. In addition, the peak shift of Ga and O in GaOOH/rGO relative to pure GaOOH is attributed to electron transfer from electron-rich rGO to GaOOH, which indicates the chemical interaction between GaOOH and rGO (Supplementary Fig. 9).

Subsequently, GaOOH/rGO catalyst has been assembled into a flow cell (electrode area: $1 \times 1$ cm$^2$) to perform the fixation. The cathode and anode chambers were fed with nitrate aqueous solutions, and the gas chamber was fed with $CO_2$ gas, respectively. Upon electrolysis, $NO_3^-$ has been electrochemically reduced to $NH_3$ in cathode chamber, where the as-generated $NH_3$ (or its ionized state of $NH_4^+$) could increase $CO_2$ dissolution. On the other hand, $CO_2$ in gas chamber enters into cathode chamber via gas diffusion electrodes, which then dissolves in electrolytes to produce $HCO_3^-$. After the reaction, it is straightforward to separate $NaHCO_3$ from other salts in the electrolytes based on their physical properties (i.e., solubility difference).

Linear scanning voltammetry (LSV) curves demonstrate the elevated current densities with negative polarization potentials (Supplementary Fig. 10), which is a common observation of cathodic reactions. This is consistent with chronopotentiometry tests showing increased current densities with more negatively applied potentials, leading to the productivities of 87.5, 156.3, 251.9, 313.6 and 374.4 mmol L$^{-1}$ h$^{-1}$ cm$^{-2}$ at 100, 200, 300, 400 and 500 mA cm$^{-2}$, respectively (Fig. 2d, e; Supplementary Figs. 11–13). During the chronopotentiometry test, we have found the initial pH of electrolyte as 6.2, which then rapidly increased to 8.3 after a few minutes. (Supplementary Fig. 14). Notably, GaOOH/rGO hybrid catalyst outperforms its individual counterparts at the same applied potentials (Fig. 2f; Supplementary Fig. 15), i.e., 162.2, 243.6 and 293.0 mmol L$^{-1}$ h$^{-1}$ cm$^{-2}$ for GaOOH/rGO, 80.4, 141.7 and 228.3 mmol L$^{-1}$ h$^{-1}$ cm$^{-2}$ for GaOOH and 67.4, 126.7 and 218.8 mmol L$^{-1}$ h$^{-1}$ cm$^{-2}$ for rGO at −1.0, −1.5, and −2.0 V vs. RHE, respectively.

We have noted that $NO_3^-$ and $CO_2$ feedstocks have played a significant role in the formation of $NaHCO_3$. Firstly, $NO_3RR$ can convert $NO_3^-$ into $NH_3$ as the dominant product with minor amounts of $NO_2^-$ and $N_2H_4$ (Supplementary Figs. 16–21). As comparison to Ar, pumping $CO_2$ into the system can accelerate $NO_3^-$-to-$NH_3$ conversion (Supplementary Fig. 22), where only trace gaseous $H_2$ and CO byproducts were detected (total Faradaic efficiencies <14.6%) with no other $CO_2RR$ products (like HCOOH; Supplementary Figs. 23–25). Meanwhile, different from previous reports[26,32,33], urea has seldom been detected during the reaction process, which indicates the fixation of $CO_2$ instead of C−N coupling (Supplementary Figs. 26, 27). Noteworthily, when we replace $CO_2$ with Ar, the flow cell system can only produce NaOH and $NH_3$ under the same experimental conditions (Supplementary Figs. 28, 29). If we replace $NaNO_3$ with $Na_2SO_4$, $NaHCO_3$ can still be produced but at compromised productivities, i.e., 63.9, 122.8, 162.8, 206.1 and 254.0 mmol L$^{-1}$ h$^{-1}$ cm$^{-2}$ at 100, 200, 300, 400 and 500 mA cm$^{-2}$, respectively (Fig. 2g). Further, the effects of $NO_3^-$ concentration and $CO_2$ partial pressure have also been examined, which shows different impacts on the system (Supplementary Figs. 30, 31).

The synergistic effect between GaOOH and rGO has been further evidenced by a series of experimental results, where the hybrid catalyst outperforms its individual components in all of the following criteria including electrochemically active surface area (ECSA; 9.48 mF cm$^{-2}$ for GaOOH/rGO vs. 3.14 mF cm$^{-2}$ for GaOOH vs. 6.35 mF cm$^{-2}$ for rGO), electrochemical impedance spectra (EIS; 19.9 Ω for GaOOH/rGO vs. 70.1 Ω for GaOOH vs. 41.5 Ω for rGO), productivities (293.0 mmol L$^{-1}$ h$^{-1}$ cm$^{-2}$ for GaOOH/rGO vs. 228.3 mmol L$^{-1}$ h$^{-1}$ cm$^{-2}$ for GaOOH vs. 218.8 mmol L$^{-1}$ h$^{-1}$ cm$^{-2}$ for rGO), energy consumption (0.582 kg kWh$^{-1}$ for GaOOH/rGO vs. 0.809 kg kWh$^{-1}$ for GaOOH vs. 1.038 kg kWh$^{-1}$ for rGO) and reaction onset potential (−0.17 V for GaOOH/rGO vs. −0.29 V for GaOOH vs.

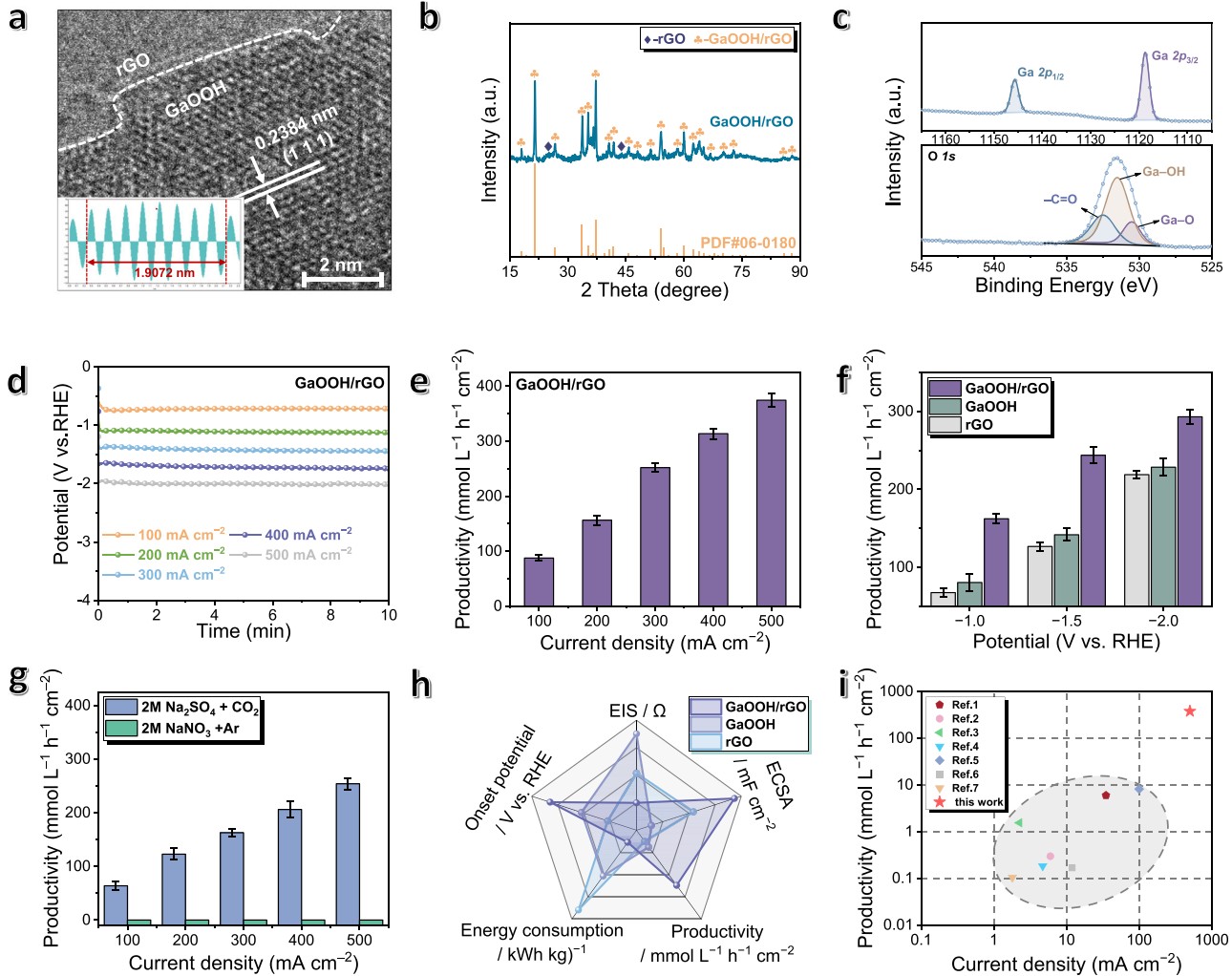

**Fig. 2 | Catalyst characterizations and electrochemical properties. a** high-resolution transmission electron microscopy (HR-TEM) images with white lines marking lattice fringes with the border between rGO and GaOOH, inset with the intensity profile of the (111) lattice plane. **b** X-ray diffraction (XRD) patterns of GaOOH/rGO. **c** The X-ray photoelectron spectroscopy (XPS) for Ga *2p* and O *1 s* in GaOOH/rGO. **d**, **e** The chronopotentiometry tests and corresponding NaHCO₃ productivities, with the error bars corresponding to the standard deviation. **f** The chronoamperometry tests and corresponding productivities of GaOOH/rGO, GaOOH and rGO catalysts, with the error bars corresponding to the standard deviation. **g** The chronopotentiometry tests for producing NaHCO₃ with error bars in 2 M Na₂SO₄ solution with CO₂ pumping and 2 M NaNO₃ solution with Ar pumping, respectively. **h** The comparison of electrochemical performances for GaOOH/rGO, GaOOH and rGO catalysts. **i** the comparison of NaHCO₃ productivities with previous reports of electrochemical methods (Please also see Supplementary Table 1). Source data for (**b**–**i**) are provided as a Source Data file.

−0.39 V for rGO; Fig. 2h; Supplementary Figs. 32, 33; Supplementary Table 3). Moreover, the GaOOH/rGO electrode exhibited durability for 100 h (Supplementary Figs. 34, 35). By comparing with the literature, the performance of our system is competitive as comparison to all previous reports pertaining to the synthesis of NaHCO₃ (Fig. 2i; Supplementary Table 1)[14,18,19,34–37].

**Mechanism study**

To understand the origin of superior electrochemical activities, mechanism study has been conducted by combining a series of experimental and theoretical characterizations, starting with X-ray absorption near edge structure spectra (XANES) showing the valence state of Ga in GaOOH/rGO in the range of 0 ~ +3 (i.e., between Ga foil and Ga₂O₃; Fig. 3a). This result is consistent with extended X-ray absorption fine structure spectra (EXAFS) and corresponding fitting results (Fig. 3b; Supplementary Figs. 36, 37; Supplementary Table 4), where GaOOH/rGO and Ga₂O₃ display comparable Ga−O shell coordination number (2.491 vs. 3.007) and bond lengths (1.948 vs. 1.873 Å). The oxidation state of Ga inside GaOOH/rGO would facilitate the

adsorption with electron-rich NO₃⁻ feedstock for favorable NO₃RR. Secondly, the interaction between GaOOH and rGO is evidenced by density functional theory (DFT) simulations showing charge transfer between Ga (GaOOH)−O (rGO)−C (rGO), which is attributable to GaOOH bridged to rGO through the oxygen functional groups on the surface of rGO (Fig. 3c). The formation of Ga−O−C can give rise to an electronic hinge structure, which endows enhanced electrical conductivity for fast charge transport.

The reaction mechanism was probed by CO₂ adsorption experiments and *operando* characterizations. Firstly, CO₂ Brunauer-Emmett-Teller (BET) experiment of GaOOH/rGO indicates a pore area of 62.2 m² g⁻¹ and CO₂ adsorption capacity of 7.98 cm³ g⁻¹, which is consistent with the ECSA results (Supplementary Fig. 32), confirming its potential for physical CO₂ adsorption (Supplementary Fig. 38). Secondly, *operando* Raman tests were employed to monitor the intermediates (Fig. 3d; Supplementary Fig. 39)[38]. A series of Raman vibration peaks have been observed: the band at 1047.8 cm⁻¹ from NO₃⁻ in the electrolyte, the band at 1552.6 cm⁻¹ from *NHO intermediate[22], the bands at 1016.9 and 1345 cm⁻¹ from

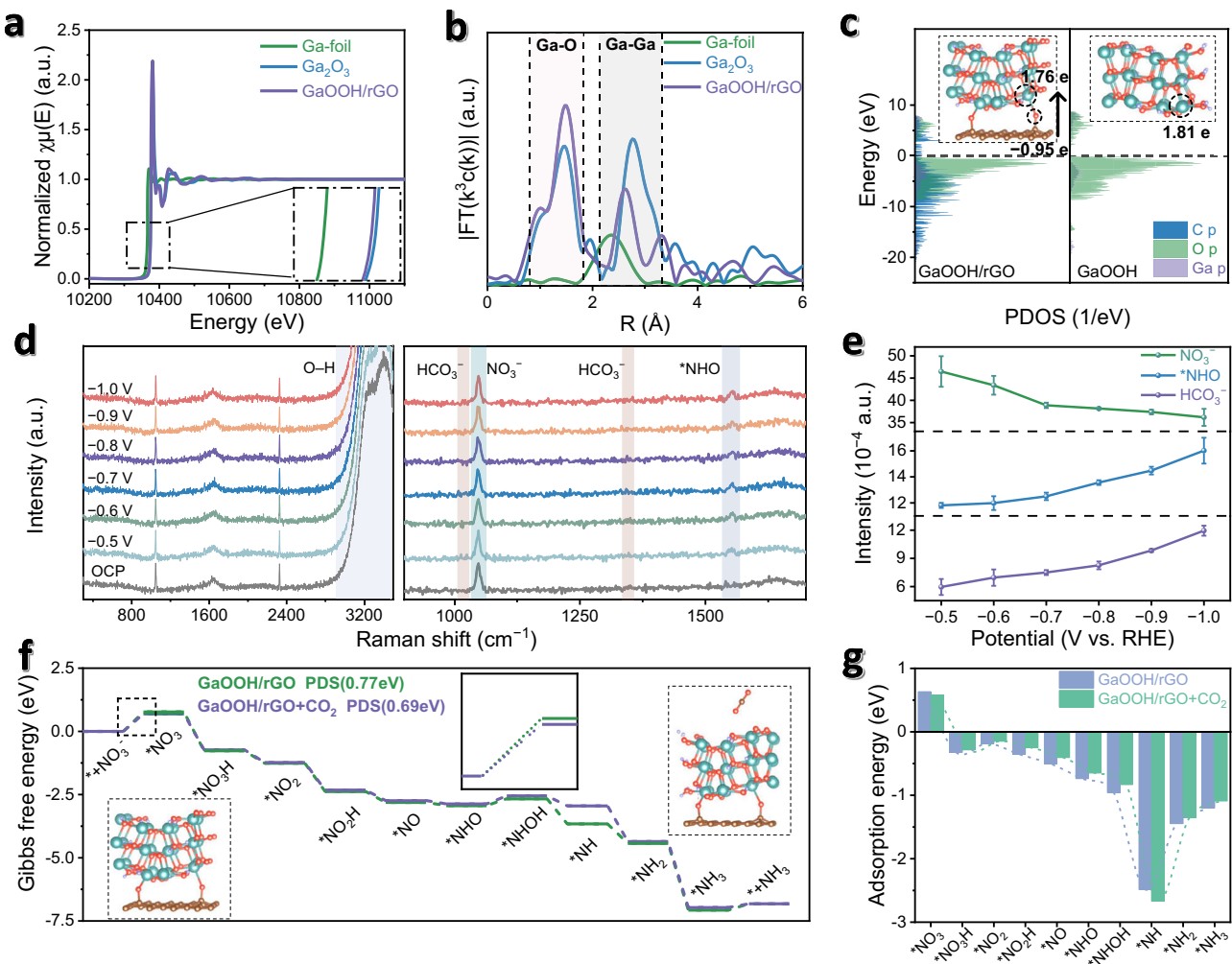

**Fig. 3 | Mechanism study. a** X-ray absorption near edge structure (XANES) spectra of GaOOH/rGO, Ga$_2$O$_3$ and Ga foils. **b** Extended X-ray absorption fine structure (EXAFS) spectra of GaOOH/rGO, Ga$_2$O$_3$ and Ga foils. **c** Density of state (DOS) profile of GaOOH/rGO and GaOOH, where the inset is the optimized structures and Bader charge transfer. **d** The *operando* Raman spectra of GaOOH/rGO under different applied potentials. **e** The relative peak intensity analyses in *operando* Raman spectra, standardized with the peak intensity of H$_2$O. The error bars correspond to the standard deviation. **f** Density functional theory (DFT) calculations for free energy diagrams of NO$_3$RR on GaOOH/rGO catalyst. **g** The adsorption energies of different reaction intermediates. Source data for (**a**–**g**) are provided as a Source Data file.

HCO$_3^-$ product[39], and the band at 2326.9 cm$^{-1}$ from the atmospheric N$_2$. Particularly, by elevating applied potentials, the relative peak intensities of these reaction-related species change correspondingly (NO$_3^-$: sharply declining from 0.0046 to 0.0036; *NHO: increasing from 0.0011 to 0.0016; HCO$_3^-$: rising from 0.0006 to 0.0012), which indicates the occurrence of NO$_3$RR to produce HCO$_3^-$ (Fig. 3e). The above result is consistent to *operando* Fourier Transform Infrared Spectroscopy (FTIR, Supplementary Fig. 40). The FTIR vibration exhibits signals at 1348, 1458 and 1560 cm$^{-1}$ corresponding to NO$_3^-$, NH$_4^+$ and *NO intermediate, respectively. The FTIR vibration peak at 1653 and 1697 cm$^{-1}$ corresponds to HCO$_3^-$, while the FTIR vibration at 2349 cm$^{-1}$ corresponds to phy-sisorbed CO$_2$. Because of CO$_2$ continuously pumped into the elec-trolyte, the FTIR vibration peak of CO$_2$ is strong. Further, the FTIR vibration peaks belonging to the catalyst itself have been identified as GaOOH (2267, 2851 and 2922 cm$^{-1}$) and rGO (C–O–H: 1118 cm$^{-1}$). With the potential increases, the FTIR peak intensities of NO$_3^-$ decrease, while the FTIR peak intensities of *NO, NH$_4^+$, and HCO$_3^-$ increase. These infrared signals reveal a tandem reaction integrating electrochemical NO$_3$RR with physisorbed CO$_2$. Notably, seldom peaks of C–N intermediates have been observed in either *operando*

Raman and FTIR spectra[40], indicating the absence of C–N coupling reaction.

Importantly, during the reaction, we have noted the positive role of physically adsorbed CO$_2$ promoting NO$_3$RR at the GaOOH/rGO electrode interface by DFT simulations. The theoretical model has been built according to experimental results comprising of GaOOH supercell with (111) plane anchored on rGO surface via oxygen-containing groups (Supplementary Figs. 41–45; Supplementary Data 1). Without CO$_2$, NO$_3^-$ reduction is a typical eight-electron-transfer process (Fig. 3f; Supplementary Figs. 46, 47)[41], starting by the first exothermic step of NO$_3^-$ adsorption to *NO$_3^-$ (0.77 eV), followed by a series of endothermic steps of *NO$_3$H (−0.77 eV), *NO$_2$ (−1.26 eV), *NO$_2$H (−2.38 eV), *NO (−2.81 eV), *NHO (−2.94 eV), *NHOH (−2.67 eV), *NH (−3.66 eV), *NH$_2$ (−4.43 eV) and *NH$_3$ (−7.07 eV), respectively. Consequently, the potential limiting step of NO$_3^-$ reduction is * → *NO$_3^-$ (0.77 eV). While in the presence of CO$_2$, the free energy barrier for * → *NO$_3^-$ has decreased to 0.69 eV, indicating its positive role in facilitating the potential-determining step.

To get a better understanding of the positive role of CO$_2$, theo-retical adsorption energies have been calculated for GaOOH/rGO to

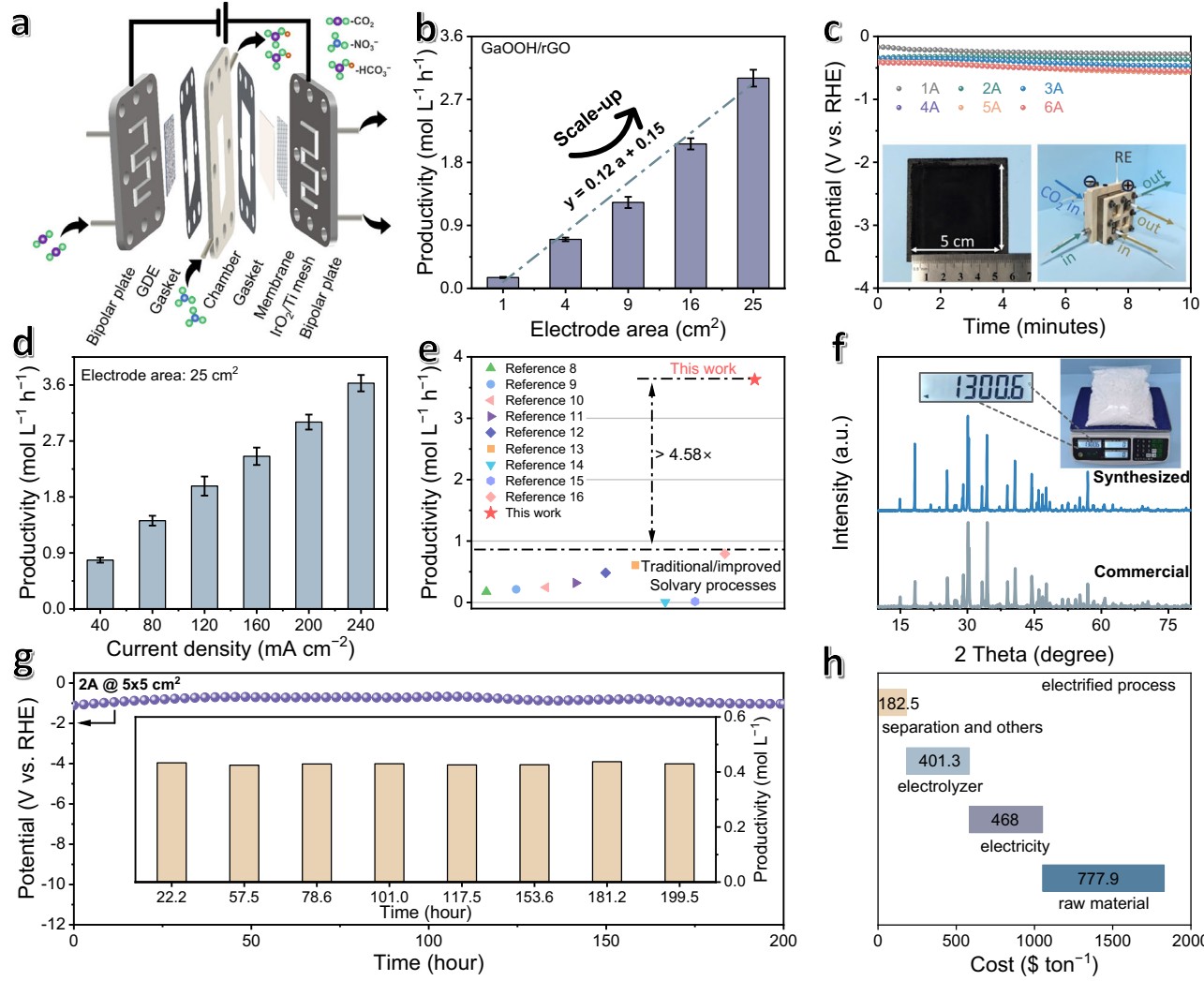

**Fig. 4 | Scaled-up production. a** the schematic diagram of the prototype device. **b** The NaHCO$_3$ productivities of GaOOH/rGO at different electrode areas, with the error bars corresponding to the standard deviation. **c** The chronopotentiometry tests of GaOOH/rGO at ampere-level currents, with the inset showing the photographs of electrodes and electrolytic cells. **d** The NaHCO$_3$ productivities of GaOOH/rGO catalysts with the electrode area of $5 \times 5$ cm$^2$ at different current densities, with the error bars corresponding to the standard deviation. **e** The comparison of NaHCO$_3$ productivities with previously reported traditional/improved Solvay processes (Please also see Supplementary Table 5). **f** The XRD patterns of the as-produced and commercial NaHCO$_3$, with the inset showing the photographs of the kilogram-level product. **g** The durability test of GaOOH/rGO at a current of 2 A, inset with the measured NaHCO$_3$ concentration during the 200-h durability test. **h** Roadmap to NaHCO$_3$ production cost by successive changes to cost-relevant parameters for our electrified process. Source data for (**b–h**) are provided as a Source Data file.

different intermediates with and without CO$_2$ (Fig. 3g). Firstly, the adsorption energy of NO$_3^-$ on GaOOH/rGO is positive, which is consistent with the non-spontaneous adsorption step as predicted by the Gibbs free energy diagram. Except for NO$_3^-$, all of the other intermediates show negative adsorption energies corresponding to exothermic steps. Interestingly, most adsorption energies of intermediates show a slight drop in the presence of CO$_2$, owing to the electron-rich CO$_2$ contributing to electrostatic induction to GaOOH/rGO, thus leading to a decreased electron density at the active sites for adsorbing intermediates (Supplementary Figs. 48–57). Further, we have examined the increased adsorption energies of *NO$_3^-$ intermediate with CO$_2$ by partial density of states (PDOS) plots, which illustrate the distribution of electrons in atomic orbitals when the active center adsorbs individual intermediates (Supplementary Figs. 58–67). For *NO$_3^-$ intermediate, the PDOS of bonding orbital is enhanced in the presence of CO$_2$, which is profitable to the stabilization and conversion of the intermediates, thus promoting the * → *NO$_3^-$ potential-determining step.

## Scaled-up production

To further evaluate its potential for practical applications, the system has been scaled up to a prototype module cell up to 25 cm$^2$ (Fig. 4a and inset in Fig. 4c). The module cell is comprised of a titanium bipolar plate with serpentine channel, silicone gaskets, hollow chambers, anion exchange membrane, gas diffusion electrodes and iridium oxide/titanium mesh. During electrochemical test, the anode electrolyte flows through the titanium bipolar plate on one side, and the membrane directly covers the anode iridium oxide/titanium mesh. While the cathode and anode are in close proximity to one another to minimize cell voltage.

We started the scale-up production by exploring the influence of different parameters on productivity (Fig. 4b–d). Firstly, with the electrode areas elevating from 1, 4, 9, 16 to 25 cm$^2$, the quantified NaHCO$_3$ productivities are 0.16, 0.70, 1.23, 2.07 and 3.00 mol L$^{-1}$ h$^{-1}$, respectively. The overall productivity shows a nearly linear correlation relationship to electrode areas (productivity = 0.12 * area + 0.15; Fig. 4b), highlighting the advantages of our device in contributing to

maximizing the use of electrode surfaces. Next, we examined the $NaHCO_3$ productivity at different electrolyte flow rates with the electrode area fixed at 25 cm$^2$ and current at 2 A (Supplementary Fig. 68). This value initializes at 1.04 mol L$^{-1}$ h$^{-1}$ at the small flow rate of 1.7 mL min$^{-1}$ owing to insufficient feedstock supply, which then increases with elevated flow rates to 1.48 mol L$^{-1}$ h$^{-1}$ at 3.4 mL min$^{-1}$, 1.58 mol L$^{-1}$ h$^{-1}$ at 5.1 mL min$^{-1}$ and reaches the plateau of 1.68 mol L$^{-1}$ h$^{-1}$ for 6.8 mL min$^{-1}$. Further increasing the flow rates to 8.5 mL min$^{-1}$ leads to seldom productivity change, indicating adequate feedstock supply for the reaction.

Next, we conducted chronopotentiometry tests to quantify the productivities at the fixed electrode area of 25 cm$^2$ (Fig. 4c). A series of negative potentials (−0.2 ~ −0.6 V vs. RHE) were applied to generate the current of 40 - 240 mA cm$^{-2}$ (i.e., 1 ~ 6 A). With the applied current densities increasing from 40, 80, 120, 160, 200 to 240 mA cm$^{-2}$ (Fig. 4d), the corresponding $NaHCO_3$ productivities are 0.79, 1.42, 1.98, 2.46, 3.00 and 3.63 mol L$^{-1}$ h$^{-1}$, respectively. Particularly at 240 mA cm$^{-2}$ (i.e., 6 A), our module cell can achieve the $NaHCO_3$ productivity of 3.63 mol L$^{-1}$ h$^{-1}$ (Fig. 4e; Supplementary Table 5)[7,12,42,43]. Beyond productivity, our electrified system demonstrates comparable performance in $CO_2$ conversion rates and energy efficiency to traditional/improved Solvay processes, positioning it as a promising candidate for large-scale industrial synthesis (Supplementary Table 6).

Besides superior activities, the potential for industrial production of our module cell has been further verified by other results. For example, the as-produced product has been collected, concentrated, and separated from the electrolytes according to solubility, which can achieve up to 1.30 kg with a purity comparable to that of commercial $NaHCO_3$ (Fig. 4f; Supplementary Figs. 69, 70; Supplementary Table 7). In addition, the module cell can sustain long-term operation for over 200 h at the current of 2 A, continuously producing 0.43 mol L$^{-1}$ of $NaHCO_3$ solution (Fig. 4g). After the test, the electrode shows little activity decay (Supplementary Figs. 71, 72) and morphology alternations (Supplementary Fig. 73). Further economic evaluations demonstrate the production cost of the proposed modular cell. Based on renewable electricity priced at $0.03 kWh$^{-1}$ [44], the operational cost of the proposed *operando* process is competitive as comparison to of traditional/improved Solvay processes ($1829.7 vs. $4106.1 ton$^{-1}$ day$^{-1}$), representing a more economically viable option (Fig. 4h; Supplementary Note 2; Supplementary Fig. 74).

## Discussions
In closing, we develop a theory-guided strategy to construct a liquid metal-derived hybrid catalyst, which tames the stumbling scaling relationship into positive synergy for *operando* alkaline generation. The overall system achieves a superior $NaHCO_3$ productivity. We anticipate that the approach of octet rule-directed *operando* chemistry would prove beneficial not only to the electrosynthesis of $NaHCO_3$ but also point a way to develop many other demanding electrified chemical and catalytic synthesis processes. Further, our unexpected observation and mechanism study of breaking the scaling relationship effect by using liquid-metal-derived catalyst sits at the intersection of chemistry, catalysis, nanomaterials and electrochemistry, which opens up enormous opportunities to explore other industrially relevant mechanisms.

## Methods
### Materials synthesis
50 mg of liquid metal Ga (99.999%, HUATAI) was dispersed into the mixed solvents of 1-dodecanethiol (2 mL; 98%, Innochem)/isopropanol (10 mL; 99.7%, Aladdin) under mild ultrasonication[45]. Next, the as-resultant Ga nanoparticles were collected, washed and dispersed in isopropanol with a concentration of 2 mg mL$^{-1}$. Further, 1.5 mL of the above Ga suspension, 2 mL of graphene oxide

suspension (GO, 3 mg mL$^{-1}$, synthesized by Hummer's method[46]) and 1.5 mL of deionized water were mixed and hydrothermally treated at 150 °C for 6100 h (Ga + 2H$_2$O + GO → GaOOH/rGO + 3H$^+$). The product was collected, washed and freeze-dried to obtain the final GaOOH/rGO catalyst. For comparison, GaOOH and rGO were obtained through the same procedure in the absence of GO and Ga, respectively.

### Characterizations
Scanning electron microscopy (SEM) was proceeded on a JSM-7800F PRIME. Transmission electron microscopy (TEM) was conducted on a JEOL JEM-F200. X-ray diffraction (XRD) was conducted on a SmartLab SE with Cu Kα radiation operating at 40 kV and 30 mA. X-ray photoelectron spectroscopy (XPS) was collected on a Thermo Scientific ESCALAB 250Xi with Al Kα X-ray source. Nuclear magnetic resonance spectrometer (NMR) was carried out at AVANCE NEO 500 MHz. Gas was collected at PANNA A91 plus gas chromatograph (GC). The UV-vis absorption spectrum was measured on Agilent Cary-60 spectrophotometer. Ga K-edge X-ray absorption near edge structure (XANES) and extended X-ray absorption fine structure (EXAFS) experiments were conducted at Shanghai Synchrotron Radiation Facility.

### Electrochemical measurements
All the electrochemical tests were performed using a potentiostat (CHI 760e). After the electrochemical experiments, the measured potentials were converted to the RHE scale at 25 °C according to the following equation: $E_{RHE} = E_{measure} + 0.059 \, pH + E^0$, where $E_{RHE}$ is the converted potential vs. RHE, $E_{measure}$ is the measured potential with respect to the reference electrode, and $E^0$ is the standard potential of Ag/AgCl at 25 °C, i.e. The reference electrode was calibrated using a reversible hydrogen electrode (RHE). Electrochemical impedance spectroscopy (EIS) was carried out in the same potentiostat in a frequency range from 0.01 Hz to 10,000 Hz with an amplitude of 10 mV. All electrochemical measurements were carried out without iR-correction. The electrochemically active surface area (ECSA) was measured from the electrochemical double-layer capacitance ($C_{dl}$) of the surface. The capacitance was determined by measuring the non-Faradaic current associated with double-layer charging from the scan rate dependence of cyclic voltammograms.

### Flow-type electrochemical cells
In general, the flow-type cell was built by coupling anodic and cathodic compartments with an anion exchange membrane (FAB−PK−130; size: 3 cm × 3 cm; thickness: 130 µm). Firstly, the as-prepared GaOOH/rGO catalyst powder was processed into catalyst ink which was prepared by mixing 5 mg of catalyst powder with 30 µL of Nafion in 220 µL/750 µL of ultrapure water/isopropanol solution. Then the dispersed ink was sprayed onto a gas diffusion electrode (GDE, working area: 1 cm$^2$) with a mass loading of 0.2 mg cm$^{-2}$. Next, the above GDE was used as cathode (in cathodic compartment), IrO$_2$-coating titanium sheet as (in anodic compartment) and Ag/AgCl electrode as reference electrode (in cathodic compartment), respectively. Particularly in the cathodic and anodic compartments, NaNO$_3$ aqueous solution (50 mL, 2 M; prepared with ultrapure water in 25 °C) was used as electrolyte and recycled with the peristaltic pump at the rate of 35 r min$^{-1}$. The CO$_2$ gas supply rate was stabilized at 10 mL min$^{-1}$ feeding into the cathodic compartment.

### Quantification of HCO$_3^-$ product
The HCO$_3^-$ concentration was detected through secondary end-point titration method. Specifically, the pH of cathode solution was carefully titrated with standard sulfuric acid (pH = 2.35) through two steps (first to the pH = 5.1, and then to 3.5), where the bicarbonate concentrations are calculated by the following Eqs. 5–7

(Supplementary Fig. 11; Please see detailed derivation of these equations in Supplementary Note 1).

$$a_1 = \frac{[HCO_3^-]_{(a_1)}([H]_2 - [H]_1)}{[H]_2 + K} \quad (5)$$

$$a_2 = \frac{[HCO_3^-]_{(a_2)}([H]_3 - [H]_1)}{[H]_3 + K} \quad (6)$$

$$[HCO_3^-] = [HCO_3^-]_{(a_1)} + [HCO_3^-]_{(a_2)} \quad (7)$$

where $a_1$ and $a_2$ are the molar equivalents of the standard acid consumed to the first and second endpoints; $[HCO_3^-]$ is the bicarbonate concentrations; $[H]_{1, 2, 3}$ are the hydrogen ion concentrations of the original solution at the first, second and endpoints, respectively; $K$ is a conditional disassociation constant of carbonic acid (i.e., $4.2 \times 10^{-7}$). The productivity of $NaHCO_3$ was calibrated according to the $Na^+/NH_4^+$ ratio in electrolyte (Supplementary Fig. 69c).

### Operando Raman spectroscopy
*Operando* Raman spectra were collected on a high-resolution Raman spectrometer equipped with an external optical path and a CHI1140C electrochemical workstation. Firstly, a catalyst ink was prepared by mixing 10 mg of catalyst with 60 μL of Nafion in 220 μL/720 μL of water/isopropanol solution, which was then sprayed onto a carbon paper substrate (2 cm × 2 cm, mass loading of 2.5 mg cm⁻²). The catalyst-coated carbon paper, Pt foil and Ag/AgCl electrode were assembled in a flow-type cell employed as working, counter and reference electrodes, respectively. All of the detected spectra were collected with a 532 nm laser wavenumber.

### Operando FTIR spectroscopy
*Operando* Fourier Transform Infrared Spectroscopy (FTIR) spectra were collected on a Bruker INVENIO instrument equipped with an accessory and a CHI1140C electrochemical workstation. Firstly, a catalyst ink was prepared by mixing 10 mg of catalyst with 60 μL of Nafion in a mixed solution of ultrapure water/isopropanol (220 μL/720 μL), which was then dripped onto a copper electrode (0.196 cm²). The catalyst-coated copper electrode, Pt foil and Ag/AgCl electrode were assembled in a flow-type cell as working, counter and reference electrodes, respectively.

### Scaled-up prototype module cell
Firstly, the GaOOH/rGO catalyst was loaded on the GDE (working area: 25 cm²) with a mass loading of 0.2 mg cm⁻². Next, the GDE was used as the cathode. $IrO_2$-coating titanium mesh and silver silk were employed as anode and reference electrode, respectively. For cathodic and anodic compartments, 50 mL 2 M $NaNO_3$ aqueous solution was used as the electrolyte and recycled with the peristaltic pump at the rate of 35 r min⁻¹. The gas supply rate was stabilized at 50 mL min⁻¹ feeding into the cathodic compartment.

### Product purification and collection
After electrolysis, the as-generated electrolyte (20 mL) contains of a mixture of $NaNO_3$, $NH_4NO_3$, $Na_2CO_3$, $NaHCO_3$, $NH_4HCO_3$ and $(NH_4)_2CO_3$. These chemicals show different physical/chemical properties, particularly solubilities in discrepant pH environments (Supplementary Table 7). Therefore, we have adjusted the pH of electrolyte to facilitate the collection of $NaHCO_3$. Experimentally, we have adjusted the pH of electrolyte to ~8.3 by sulfuric acid. The as-formed solution was then evaporated at 35 °C until the onset of $NaHCO_3$ crystallization (~5100 h). Immediately, the solution was frozen for 2100 h, where the as-precipitated white crystal is the target product of $NaHCO_3$.

### Energy consumption calculation
The energy consumption represents the amount of energy required to form a product per unit mass in the current system[13]. So, the energy consumption ($E_c$, kWh/kg $NaHCO_3$) for producing $NaHCO_3$ was calculated according to the equation:

$$E_c = \frac{\int_0^t U \times I(t)dt}{C \times V \times M} \quad (8)$$

Where $U$ represents the electrolysis voltage, and $I$ is the reaction current. $C$ means the production concentration in the electrolyte, $V$ is the volume of electrolyte, and $M$ is the molar mass of $NaHCO_3$.

### Computational methods
DFT calculations were performed by MedeA-Vienna Ab initio Simulation Package (VASP). The Perdew-Burke-Ernzerhof (PBE) generalized gradient approach was used to define the exchange-correlation potential[47,48]. The interaction between atomic cores and electrons was described by using the projector augmented wave method (PAW)[49,50]. The plane wave energy cutoff was set to be 500 eV. The convergence criterion was set to be $10^{-5}$ eV and 0.04 eV/Å for energy and force in the geometry optimizations, respectively. The Brillouin zone in the real space was sampled with a 2 × 2 × 1 Monkhorst-Pack K-point grid. Hubbard-U correction method (DFT + U) was carried out to improve the description of highly correlated Ga 3 d orbitals with the value of U−J set to be 2.5 eV. A Gaussian smearing method was employed with 0.05 eV width.

The detailed Gibbs free energy has been calculated according to the following equation:

$$G = E + ZTE - TS \quad (9)$$

Where $G$, $E$ and $ZTE$ refer to chemical Gibbs free energy, electronic energy and zero-point energy, respectively. The entropy can be calculated by the sum of the vibrational, rotational, translational, and electronic contribution as to:

$$S = S_v + S_r + S_t + S_e \quad (10)$$

For the case of solids and adsorbates, some approximations can be adopted: Translational and rotational motions can be omitted, therefore, $S_t \approx 0$ and $S_r \approx 0$. Since $S_e \approx 0$ at the fundamental electronic level. In this case, all the entropy values come from the vibrational contribution: $S = S_v$

Finally, Gibbs free energy for different states was calculated as to:

$$G = E + ZTE - TS_v \quad (11)$$

### Determination of products from nitrate reduction reaction (NO₃RR)
The concentration of $NH_3$ byproduct was determined by indophenol blue method with modification[51]. Initially, a portion of electrolyte was extracted from the electrolytic cell and subsequently diluted to the detection range. Next, 2 mL of the diluted electrolyte was mixed with NaOH solution (2 mL, 1 M) containing of 5 wt% salicylic acid and 5 wt% sodium citrate, followed by the addition of 1 mL of 0.05 M NaClO and 0.2 mL of 1.0 wt% sodium nitroferricyanide solution. After 2 h under ambient condition, the absorption peak was measured by UV-vis spectrophotometer at the wavelength of 655 nm.

The concentration of $NO_2^-$ byproduct was determined by UV-vis spectrophotometer[51]. Firstly, a color reagent was prepared as follows: 4 g of p-aminobenzenesulfonamide, 0.2 g of N-(1-naphthyl) ethylenediamine dihydrochloride and 10 mL of phosphoric acid were added into 50 mL of deionized water and mixed thoroughly. Subsequently,

5 mL of the diluted electrolyte and 0.1 mL of the color reagent were mixed for 20 min under ambient condition. The absorption spectrum was measured by UV-vis spectrophotometer at the wavelength of 540 nm.

The concentration of $N_2H_4$ byproduct was determined by Watt and Chrisp method[52]. The color reagent was prepared by 5.99 g of para-(dimethylamino)benzaldehyde, 30 mL of concentrated hydrochloric acid and 300 mL of ethanol. Subsequently, 5 mL of the diluted electrolyte and 5 mL of color reagent were mixed for 10 min under ambient condition. The absorption spectrum was measured by UV-vis spectrophotometer at the wavelength of 457 nm.

### Electrocatalytic $CO_2$ reduction reaction ($CO_2$RR) activities

The $CO_2$RR was conducted in a traditional three-electrode flow cell system comprising an $IrO_2$/Ti counter electrode, an Ag/AgCl reference electrode and the as-synthesized GaOOH/rGO, GaOOH, rGO catalyst as the working electrodes. The electrolyte was 2 M $NaNO_3$ aqueous solution. We collected the gas products by airbags, which were detected through a gas chromatograph (GC) equipped with a thermal conductivity detector and a flame ionization detector. The Ar (99.99%) acted as carrier gas. The flow rate of $CO_2$ was est to be 80 ml min$^{-1}$ with flow controller. On the other hand, the liquid products were measured by nuclear magnetic resonance (NMR) spectroscopy (500 M). Typically, 700 μL of electrolyte is firstly mixed with 30 μL of $D_2O$ and dimethyl sulfoxide (DMSO) mixed solution.

The Faradaic efficiency (FE) of different products was calculated as follows[53]:

$$FE = \frac{c \times V \times n \times F}{I \times t} \times 100\% \qquad (12)$$

Here, $c$ represents the concentration of products; $V$ represents the total volume, which is 300 mL for gas and 50 mL for liquid, respectively; $n$ is the number of transferred electrons, which is 2 for both CO, $H_2$ and formate; $F$ is the Faradaic constant (96485 C mol$^{-1}$); $t$ is reaction duration, $I$ is the applied steady current.

### Possible urea product from carbon–nitrogen (C–N) coupling

The concentration of possible urea product was determined by the diacetyloxime method[54]. Specifically, 500 mg of diacetylmonooxime (DAMO) and 10 mg of thiourea (TSC) were dissolved in 100 mL of distilled water, resulting in a solution designated as DAMO–TSC. On the other hand, 10 mL of concentrated phosphoric acid and 30 mL of concentrated sulfuric acid were added to 60 mL of distilled water, into which 10 mg of $FeCl_3$ was added with vigorous mixing, resulting in the acid–iron solution.

To determine the possible urea product, 1 mL of electrolyte was extracted from the cathode compartment, which was then mixed with 1 mL of the DAMO–TSC solution and 2 mL of the ferric acid solution, respectively. Next, the aforementioned solution was heated to 100 °C for 50 min. After cooling down to 25 °C, the absorbance was measured by UV-vis spectrophotometer at the peak position of 520 nm.

## Data availability

The datasets generated and analysed during the present study are included in the main text and Supplementary Information. Source data are provided with this paper.

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

## Acknowledgements

This work was financially supported by National Natural Science Foundation (Grant No. 52488201, 92163124 and 52376193), Jiangsu Natural Science Foundation (Grant No. BK 20230097) and European Union's Framework Program for Research and Innovation Horizon 2020 (2014–2021). The authors also acknowledge the BL20U1, BL17B and BL14W1 beamlines at the Shanghai Synchrotron Radiation Facility.

## Author contributions

S.C. supervised the project and designed the experiments. Q.H. and J.J.D. performed experiments and DFT calculations. S.C., Q.H., J.J.D. and M.A. discussed the results for paper preparation.

## Funding

## Competing interests

Sheng Chen has filed Chinese provisional patent applications (No. 2025103278534) based on this work. Other authors declare no competing interests.
