## [Transparent Peer Review file · Nature Communications]

Operando-electrified Solvay process with productivity overtaking benchmark industrial route

Corresponding Author: Professor sheng Chen

Version 0:

Reviewer comments:

Reviewer #1

(Remarks to the Author)

This work presents an innovative operando-electrified system integrating eight-electron nitrogen chemistry into the Solvay process, achieving a remarkable NaHCO_3 productivity of $3.69 \text{ mol L}^{-1} \text{ h}^{-1}$. The design of a GaOOH/rGO catalyst to suppress parasitic reactions (CO_2RR , HER, CNR) and enhance NO_3RR -driven alkaline generation is commendable. While the study demonstrates significant progress in green chemical synthesis, several critical issues require clarification to strengthen mechanistic insights and justify the claimed process innovation.

1. Experimental Evidence Requires Supplementation

The mechanism of CO_2 physical adsorption promoting NO_3RR lacks direct experimental evidence (e.g., in situ FTIR or XPS to confirm CO_2 adsorption states). While DFT simulations and operando Raman spectroscopy suggest CO_2 enrichment at the electrode surface, no data explicitly demonstrate CO_2 adsorption behavior or quantify its coverage under reaction conditions. Perhaps authors could perform CO_2 temperature-programmed desorption (CO_2 -TPD) to characterize the adsorption capacity and strength of CO_2 on GaOOH/rGO . And also conduct operando XPS or in situ FTIR to identify the chemical state of adsorbed CO_2 (e.g., physisorbed vs. chemisorbed) and its dynamic changes during electrolysis..

2. Structural Focus Adjustment

While the introduction emphasizes the innovation in replacing the Solvay process with an operando-electrified system, the main text overly focuses on material characterization (e.g., SEM, XRD) and DFT calculations, diluting the discussion on process-level advancements. Please reorganize the Results/Discussion section to highlight process-level advantages, for example:

a) Compare reaction rates, selectivity, and energy efficiency between the operando-electrified system and traditional Solvay process in a dedicated table.

b) Integrate economic analysis (Supplementary Fig. 48) into the main text to strengthen the case for commercialization.

3. Quantification of CO_2 Promotion Mechanism

The DFT results show that CO_2 reduces the energy barrier of the NO_3^- adsorption step from 0.77 eV to 0.69 eV. However, the ratio of CO_2 -occupied active sites or its adsorption coverage under operational conditions remains unclear. It's better to vary CO_2 partial pressure and NO_3^- concentration to map the correlation between $\text{CO}_2/\text{NO}_3^-$ ratio and HCO_3^- productivity.

4. Strengths and Innovation

Reorganize the Introduction to emphasize process innovation over material-centric descriptions. Meanwhile please give a more detailed introduction of the development of GaOOH/rGO catalyst design.

Reviewer #2

(Remarks to the Author)

The manuscript authored by Chen et al. reports an operando-electrified system of delivering two environmental pollutants (CO_2 and NO_3^-) into an electrochemical cell to produce NaHCO_3 by employing a liquid metal-derived hybrid catalyst (GaOOH/rGO). The overall system achieves the maximum NaHCO_3 productivity up to $3.69 \text{ mol L}^{-1} \text{ h}^{-1}$. Overall, the manuscript is logically organized, with its mechanistic interpretation largely supported by both experimental and computational evidence. However, some issues are still in existence in this study. I think a further revision is required for the manuscript to improve its quality.

- (1) It is claimed that NO_3^- is converted into NH_3 during the NO_3RR process, which is also a value-added product. It would be better if the authors could further evaluate the faradaic efficiency and yield rate for NH_3 generation.
- (2) The NO_3^- concentration in the electrolyte is 2 M. The reviewer wonders whether the introduced NO_3^- concentration has influence on the concentration of local alkaline and the productivity of NaHCO_3 .
- (3) All electrochemical experiments should be repeated at least three times to ensure the accuracy of results. And the error bars should be added.
- (4) What is pH value for the electrolyte before testing? And how about that during the operando-electrified process?
- (5) In Figure 2b, it would be better to use the balls of the same color to represent the same atoms.

Reviewer #3

(Remarks to the Author)

This work aims to replace the traditional, energy- and cost-intensive Solvay process for NaHCO_3 production with an innovative operando-electrified synthesis approach. Inspired by the octet rule, the authors propose integrating eight-electron nitrogen chemistry to enhance local alkaline generation and CO_2 conversion efficiency. The system reportedly achieves a productivity of $3.69 \text{ mol L}^{-1} \text{ h}^{-1}$, which is claimed to exceed the benchmark performance of the Solvay process and surpass existing electrosynthetic methods. While the manuscript presents an interesting and potentially impactful concept, several fundamental issues limit its suitability for publication in its current form. Therefore, I suggest that it be considered for publication after further revision.

First, although the reported productivity is emphasized as a key highlight, the supporting evidence lacks sufficient rigor and persuasiveness. The experimental design and data interpretation do not convincingly establish the claimed enhancement over industrial benchmarks, and the methodology for productivity quantification requires greater clarity and validation.

Second, the manuscript suffers from notable issues in logical flow and coherence. Besides, several sections of the manuscript present incomplete or irrelevant data interpretation. More comprehensive mechanistic elucidation and a clearer structure would be necessary to support the high-level claims made. The following are some of the specific reasons :

1. The formatting of the figures in the main text is inconsistent. For example, some figures are not enclosed or properly framed. Please standardize the figure formatting throughout the manuscript.
2. Language and grammar issues: Phrases such as "challenge problem," "Other than hydrogen chemistry," and "maximumly eight-electron chemical reactions" are grammatically incorrect or awkwardly phrased. "Despite of" should be corrected to "despite." "Maximumly" should be "maximally"; "owning to" should be owing to. A thorough language revision is necessary to correct grammar and improve readability.
3. Define terms like "operando nitrogen chemistry" and "scaling relationship" more precisely.
4. The introduction does not fully meet the norms and logical expectations of scientific writing. The final paragraph of the Introduction should clearly state the research objective, briefly introduce the core strategy or hypothesis, and highlight the innovation and significance without going too deep into technical details. The current version reads more like a mix of Introduction and Results, making it less effective in providing a concise transition into the body of the paper. Consider streamlining the content, correcting grammatical issues, and restructuring the paragraph to improve clarity, coherence, and academic tone. Major issues: too much technical detail for an introduction; language and grammar issues; paragraphs jump between multiple ideas with insufficient transitions.
5. How to prove the strong interaction between GaOOH nanoparticles and rGO .
6. For the completeness of the article, to attract more attention and to increase the readership, the introduction should cite some of the latest advances in NO_3RR and CO_2RR , such as: *Nat Commun.* 2024, 15, 3524; *Nat Commun.* 2025, 16, 731; *J. Am. Chem. Soc.* 2025, 147, 8871–8880.

Reviewer #4

(Remarks to the Author)

In this manuscript, the authors present an operando-electrified Solvay process that utilizes the electrochemical NO_3^- reduction reaction to create a locally alkaline micro-environment, facilitating the CO_2 -to- HCO_3^- conversion. While the productivity of NaHCO_3 is impressive, it appears highly challenging for this operando-electrified Solvay process to replace the traditional Solvay process. In my view, although the performance is noteworthy, the conceptual innovation of this study is somewhat limited. Major concerns that arose during the review of this manuscript included:

1. (Bi)carbonate deposition represents a persistent challenge in alkaline/neutral CO_2 electrolysis, significantly compromising both system efficiency and operational durability. This issue not only reduces the CO_2 single-pass conversion efficiency but also blocks the gas diffusion pathway in the gas diffusion electrode (GDE), ultimately deteriorating the CO_2RR performance. In this work, NO_3RR is employed to generate a highly alkaline micro-environment, facilitating the CO_2 -to- HCO_3^- conversion, while the GDE is utilized to enhance CO_2 mass transport. However, this design raises concerns regarding potential issues analogous to those encountered in alkaline/neutral CO_2 electrolysis. Specifically, will (bi)carbonate deposit on the GDE surface and within its gas channel? If so, could (bi)carbonate precipitate hinder the diffusion of CO_2 and thereby reduce the productivity of NaHCO_3 ? Will this problem affect the performance of the GaOOH/rGO catalyst and the lifespan of the entire system? Potential solutions to mitigate (bi)carbonate deposition should be discussed. Furthermore, although the authors reported a 200-hour stability of this system, it remains unclear whether the system exhibited performance degradation over time. A detailed analysis of performance decline is essential for identifying failure mechanisms, which in turn would inform the design and optimization of this system for practical applications.
2. What are the NO_3RR performances of the GaOOH/rGO , GaOOH , and rGO catalysts? A closed electron balance should be given.
3. The quality of the in situ Raman spectra in Figure 3d is too poor to draw definitive conclusions about the presence of

HCO₃⁻. Moreover, the intensity of the Raman signal is strongly influenced by the test parameters and reaction environment (e.g., bubbles). Therefore, directly comparing the Raman signal intensity is unreasonable. It is recommended to introduce a stable internal standard for the Raman test to improve reliability.

4. In line 168, the authors highlight the charge transfer between Ga and C based on DFT calculations. Can this conclusion be further supported experimentally by XPS and XAS data? Comparing the XPS and XAS spectra of GaOOH and GaOOH/rGO catalysts would help clarify the direction of electron transfer and confirm the interaction between GaOOH and rGO.

5. Detailed characterization of the GaOOH/rGO and GaOOH catalysts, both during and after NO₃RR, is necessary to determine the true structure of catalysts under reduction potential. The applied reduction potential ranges from -0.7 V~2 V vs RHE, it is essential to address whether GaOOH will be reduced to metallic Ga under these conditions. The DFT calculation model should align with the actual structure of the catalysts under operating conditions, rather than the pristine GaOOH structure.

6. The discussion of the DFT calculations regarding the difference charge density (line 202) is confusing. First, the active site of GaOOH/rGO must be explicitly defined. Secondly, if electron-rich CO₂ contributes electrons to GaOOH/rGO, will this lead to an increased electron density of GaOOH/rGO, thereby hindering the adsorption of negatively charged NO₃⁻? The authors claim that physically adsorbed CO₂ promotes the NO₃RR at the GaOOH/rGO catalyst based on DFT simulations. Can the same enhancement in selectivity or activity for NO₃RR be experimentally observed on the GaOOH/rGO catalyst?

7. Are there any by-products, such as Na₂CO₃, generated during the process?

8. While this work highlights the advantages of producing NaHCO₃ in productivity, industrialization remains constrained by other factors, such as system stability. Could the authors comment on the industrialization prospects of this operando-electrified Solvay process?

Reviewer #5

(Remarks to the Author)
see attached

Version 1:

Reviewer comments:

Reviewer #1

(Remarks to the Author)

The authors have made substantial enhancements to the manuscript, particularly in emphasizing process innovation and validating CO₂ adsorption mechanisms through new in situ characterizations. These revisions significantly strengthen the work's industrial relevance and mechanistic credibility, aligning perfectly with previous review suggestions. Authors can consider the following minor remaining issues:

1. Clarify Current Density Units

Standardize current density in Fig. 4d (currently uses "A" without area specification).

2. Theoretical Justification

Cite the octet rule principle in the Introduction.

3. Economic Assumptions

Briefly note the \$0.03/kWh electricity cost rationale in the main text (currently only in SI).

Reviewer #2

(Remarks to the Author)

All major issues have been satisfactorily resolved. The manuscript is now suitable for publication.

Reviewer #3

(Remarks to the Author)

The author has resolved all the issues. This manuscript is acceptable and no further revisions are needed.

Reviewer #4

(Remarks to the Author)

The authors have revised the manuscript accordingly, and I recommend its acceptance for publication.

Reviewer #5

(Remarks to the Author)

Version 2:

Reviewer comments:

Reviewer #1

(Remarks to the Author)

Overall, the authors have addressed the previous comments effectively. I recommend this manuscript for publication.

Reviewer #5

(Remarks to the Author)

We would like to thank the Editor's invitation to revise the manuscript. We are also appreciated of the reviewer #1 for his/her constructive comments to improve the quality of this work.

After receiving the reviewer's comments, we have spent several intensive weeks in order to answer the concerns raised by the astute reviewer. We believe we have shifted the work to the next level of proof and gained a much deeper understanding of the system. Please see the following point-by-point response to the reviewer #1's comments. Thank you and best wishes,

Kind regards

The authors

Response to Reviewer #1:

Original Comment: *This work presents an innovative operando-electrified system integrating eight-electron nitrogen chemistry into the Solvay process, achieving a remarkable NaHCO_3 productivity of $3.69 \text{ mol L}^{-1} \text{ h}^{-1}$. The design of a GaOOH/rGO catalyst to suppress parasitic reactions (CO_2RR , HER, CNR) and enhance NO_3RR -driven alkaline generation is commendable. While the study demonstrates significant progress in green chemical synthesis, several critical issues require clarification to strengthen mechanistic insights and justify the claimed process innovation.*

Original Comment 1. *Experimental Evidence Requires Supplementation. The mechanism of CO_2 physical adsorption promoting NO_3RR lacks direct experimental evidence (e.g., in situ FTIR or XPS to confirm CO_2 adsorption states). While DFT simulations and operando Raman spectroscopy suggest CO_2 enrichment at the electrode surface, no data explicitly demonstrate CO_2 adsorption behavior or quantify its coverage under reaction conditions. Perhaps authors could perform CO_2 temperature-programmed desorption ($\text{CO}_2\text{-TPD}$) to characterize the adsorption capacity and strength of CO_2 on GaOOH/rGO . And also conduct operando XPS or in situ FTIR to identify the chemical state of adsorbed CO_2 (e.g., physisorbed vs. chemisorbed) and its dynamic changes during electrolysis.*

Response: We would like to thank the reviewer for his/her constructive comments. We have added relevant characterizations, and given the response as follows:

1) Following the reviewer's suggestion, we have attempted to conduct the experiment of CO_2 temperature-programmed desorption ($\text{CO}_2\text{-TPD}$), but failed to get an accurate result because of the intrinsic instability of GaOOH/rGO under temperature-programmed heating process (Please see thermal gravimetric TG analysis data, Fig. R1). Consequently, we have used another two experiments to characterize the CO_2 adsorption capacity and strength of GaOOH/rGO . Firstly, the CO_2 Brunauer-Emmett-Teller (BET) experiment was conducted (Supplementary Fig. 38), which indicates GaOOH/rGO with a pore area of $62.2 \text{ m}^2 \text{ g}^{-1}$ and a CO_2 adsorption capacity of $7.98 \text{ cm}^3 \text{ g}^{-1}$. Secondly, the electrochemical active surface area (ECSA) of

GaOOH/rGO, GaOOH and rGO were measured (Supplementary Fig. 32), which shows the highest double-layer capacity of 9.48 mF cm^{-2} for GaOOH/rGO, exceeding those of GaOOH (3.14 mF cm^{-2}) and rGO (6.35 mF cm^{-2}). These results indicate that GaOOH/rGO exhibits enhanced capacity of adsorbing CO_2 for electrochemical reactions.

3) Next, *operando* FTIR experiments on the electrode surface have been tested under polarized potentials (Supplementary Fig. 40). These vibration peaks at 1348 cm^{-1} , 1458 cm^{-1} and 1560 cm^{-1} correspond to NO_3^- , NH_4^+ and *NO intermediate, respectively. These vibration peaks at 1653 cm^{-1} and 1697 cm^{-1} correspond to the product HCO_3^- , while the vibration at 2349 cm^{-1} corresponds to physisorbed CO_2 . Since CO_2 is continuously pumped into the electrolyte, the infrared vibration peak of CO_2 is strong. Further, the vibration peaks belonging to the catalyst itself were identified as GaOOH (2267 cm^{-1} , 2851 cm^{-1} and 2922 cm^{-1}) and rGO (C–O–H: 1118 cm^{-1}). As the potential increases, the peak intensity of NO_3^- decreases, while the peak intensities of *NO , NH_4^+ , and HCO_3^- increase. These changes in the *operando* FTIR vibration peaks of GaOOH/rGO signify its stabilization following activation at a low potential, indicating its persistent stability. The infrared signals of these species reveal a tandem system integrating electrochemical NO_3RR with physisorbed CO_2 , thus providing unambiguous evidence for the proposed reaction mechanism.

Based on the above experiments and discussions, we have added/revise the following sentences and figures to the revised manuscript.

Fig. R1. The thermal gravimetric (TG) analysis of GaOOH/rGO.

“The reaction mechanism was probed by CO₂ adsorption experiments and *operando* characterizations. Firstly, CO₂ Brunauer-Emmett-Teller (BET) experiment of GaOOH/rGO indicates a pore area of 62.2 m² g⁻¹ and CO₂ adsorption capacity of 7.98 cm³ g⁻¹, which is consistent with the ECSA results (Supplementary Fig. 32), confirming its potential for physical CO₂ adsorption (Supplementary Fig. 38)” (Page 10, line 4–Page 10, line 8)

“The above result is consistent with *operando* FTIR (Supplementary Fig. 40). The FTIR vibration exhibits signals at 1348 cm⁻¹, 1458 cm⁻¹ and 1560 cm⁻¹ corresponding to NO₃⁻, NH₄⁺ and *NO intermediate, respectively. The FTIR vibration peak at 1653 cm⁻¹ and 1697 cm⁻¹ corresponds to HCO₃⁻, while the FTIR vibration at 2349 cm⁻¹ corresponds to physisorbed CO₂. Because of CO₂ continuously pumped into the electrolyte, the FTIR vibration peak of CO₂ is strong. Further, the FTIR vibration peaks belonging to the catalyst itself have been identified as GaOOH (2267 cm⁻¹, 2851 cm⁻¹ and 2922 cm⁻¹) and rGO (C–O–H: 1118 cm⁻¹). With the potential increases, the FTIR peak intensities of NO₃⁻ decrease, while the FTIR peak intensities of *NO, NH₄⁺, and HCO₃⁻ increase. These infrared signals reveal a tandem reaction integrating electrochemical NO₃RR with physisorbed CO₂. Notably, seldom peaks of C–N intermediates have been observed in either *operando* Raman and FTIR spectra,⁴⁰ indicating the absence of C–N coupling reaction.” (Page 10, line 15–Page 11, line 2)

Supplementary Fig. 32. Effective electrochemical active surface area tests (ECSA) by CVs at different scan rates for **a**, GaOOH/rGO. **b**, GaOOH. and **c**, rGO, respectively. **d**, Linear fit results for electrochemical double-layer capacity (C_{dl}) of GaOOH/rGO, GaOOH and rGO. The experiments demonstrated that GaOOH/rGO has exhibited the highest double-layer capacity of 9.48 mF cm^{-2} , which exceeded both GaOOH (3.14 mF cm^{-2}) and rGO (6.35 mF cm^{-2}). Under electrochemical conditions, the highest ECSA of GaOOH/rGO was conducive to most active sites. This observation indicates that GaOOH/rGO has the capacity of adsorbing more CO_2 .

Supplementary Fig. 38. The BET test of GaOOH/rGO in CO₂ atmosphere. The BET results indicate the GaOOH/rGO with a pore area of 62.2 m² g⁻¹ and a CO₂ adsorption capacity of 7.98 cm³ g⁻¹. This result indicates GaOOH/rGO exhibiting good CO₂ adsorption capabilities.

Supplementary Fig. 40. The *operando* FTIR test of GaOOH/rGO catalyst at a potential range of -0.6 V ~ -1.1 V vs. RHE.

Original Comment 2. *Structural Focus Adjustment.* While the introduction emphasizes the innovation in replacing the Solvay process with an operando-electrified system, the main text overly focuses on material characterization (e.g., SEM, XRD) and DFT calculations, diluting the discussion on process-level advancements. Please reorganize the Results/Discussion section to highlight process-level advantages, for example:

a) Compare reaction rates, selectivity, and energy efficiency between the operando-electrified system and traditional Solvay process in a dedicated table.

b) Integrate economic analysis (Supplementary Fig. 48) into the main text to strengthen the case for commercialization.

Response: Thanks for your kind reminding. We have reorganized the Results/Discussion section and given the response as follows:

1) We have re-examined the experimental data and reviewed the literature, comparing reaction rates, CO₂ selectivity and energy efficiency between our operando-electrified system with the traditional Solvay method. Please see Supplementary Table 6 and relevant discussions in Page 13, line 4–Page 13, line 9.

2) We have moved the data of economic analysis (Supplementary Fig. 48 in the original paper) into the main text (Figure 4h), and given relevant discussions in Page 13, line 17–Page 13, line 19.

“Particularly at 6 A, our module cell can achieve the NaHCO₃ productivity of 3.63 mol L⁻¹ h⁻¹, which even surpasses by 4.58 times of traditional/improved Solvay processes reported in the literature (Fig. 4e; Supplementary Table 5).^{6, 11, 38, 39} Beyond productivity, our electrified system demonstrates comparable performance in CO₂ conversion rates and energy efficiency to traditional/improved Solvay processes, positioning it as a promising candidate for large-scale industrial synthesis (Supplementary Table 6).” (Page 13, line 4–Page 13, line 9)

“Further economic evaluations demonstrate that the production cost of the proposed module cell is much lower than that of the traditional/improved Solvay processes (1829.7 vs. 4106.1 \$ ton⁻¹ day⁻¹), representing a more economically viable option (Fig. 4h; Supplementary Fig. 74).” (Page 13, line 17–Page 13, line 19)

Supplementary Table 6. The comparisons of reaction rates, CO₂ selectivity and energy efficiency between our *operando*-electrified system and the traditional Solvay method.^{1, 114, 115}

Methods	Reaction rates / mol L ⁻¹ h ⁻¹	CO ₂ selectivity / %	Energy efficiency / ton NaHCO ₃ GJ ⁻¹
operando -electrified system	3.63	~ 70	0.16
traditional Solvay process	0.79 (Korean J. Chem. Eng. 41, 2163–2172 (2024).)	~ 83 (J. Clean. Prod. 468, 143087 (2024).)	0.10 (Ind. Eng. Chem. Res. 58, 3450–3458 (2019).)

The conversion rate of CO₂ (or CO₂ selectivity) is 70% under the conditions of 500 mA cm⁻² & 10 mL min⁻¹ CO₂, which is calculated as follows:

$$\text{CO}_2 \text{ conversion rate} = \frac{C \times V}{r \times t / V_m} \quad (56)$$

Where C represents the production concentration in the electrolyte, V means the volume of electrolyte, r and t refer to gas flow rate and reaction time, respectively. And V_m is the molar volume of gas (22.4 L mol⁻¹ under standard conditions). According to the above conditions, the values of C, V, r and t are 374.4 mmol L⁻¹, 50 mL, 10 mL min⁻¹ and 1 hour, respectively. The final conversion rate of CO₂ is calculated to be 70%.

Figure 4h. Roadmap to NaHCO₃ production cost by successive changes to cost-relevant parameters for our electro-synthetic process

Original Comment 3. *Quantification of CO₂ Promotion Mechanism. The DFT results show that CO₂ reduces the energy barrier of the NO₃⁻ adsorption step from 0.77 eV to 0.69 eV. However, the ratio of CO₂-occupied active sites or its adsorption coverage under operational conditions remains unclear. It's better to vary CO₂ partial pressure and NO₃⁻ concentration to map the correlation between CO₂/NO₃⁻ ratio and HCO₃⁻ productivity.*

Response: Thanks for your kind reminding. Yes, these experiments are very important for verifying mechanisms. Thus, we have conducted these experiments as follows (Supplementary Fig. 30–31):

Firstly, according to Supplementary Fig. 30, the amount of the product NaHCO₃ is closely related to the amount of NO₃⁻. That is, when the NO₃⁻ concentration is below 2 M, a higher NO₃⁻ concentration leads to a higher NaHCO₃ productivity (86.7 mmol L⁻¹ cm⁻² h⁻¹ in 0.5 M NaNO₃; 134.3 mmol L⁻¹ cm⁻² h⁻¹ in 1 M NaNO₃). When the NO₃⁻ concentration is more than 2 M, the increase rate of NaHCO₃ productivity becomes slow (293.9 mmol L⁻¹ cm⁻² h⁻¹ in 2 M NaNO₃; 302.7 mmol L⁻¹ cm⁻² h⁻¹ in 3 M NaNO₃; 321.7 mmol L⁻¹ cm⁻² h⁻¹ in 4 M NaNO₃).

On the other hand, according to Supplementary Fig. 31, when the CO₂ partial pressures range from 15% to 100%, the productivity of NaHCO₃ is comparable to that at 293.9 mmol L⁻¹ cm⁻² h⁻¹. Therefore, the partial pressure of CO₂ does not have a significant effect on the amount of NaHCO₃ produced.

“Further, the effects of NO₃⁻ concentration and CO₂ partial pressure have also been examined, which shows different impacts on the system (Supplementary Fig. 30–31).”

(Page 8, line 23–Page 9, line 1)

Supplementary Fig. 30. The productivity of NaHCO₃ for GaOOH/rGO catalyst in different NaNO₃ concentrations at 2 V vs. RHE. The amount of the product NaHCO₃ is closely related to the amount of NO₃⁻. That is, when the NO₃⁻ concentration is below 2 M, a higher NO₃⁻ concentration leads to an increase in NaHCO₃ productivity (86.7 mmol L⁻¹ cm⁻² h⁻¹ in 0.5 M NaNO₃; 134.3 mmol L⁻¹ cm⁻² h⁻¹ in 1 M NaNO₃). When the NO₃⁻ concentration is more than 2 M, the increase rate of NaHCO₃ productivity becomes slow (293.9 mmol L⁻¹ cm⁻² h⁻¹ in 2 M NaNO₃; 302.7 mmol L⁻¹ cm⁻² h⁻¹ in 3 M NaNO₃; 321.7 mmol L⁻¹ cm⁻² h⁻¹ in 4 M NaNO₃).

Supplementary Fig. 31. The productivity of NaHCO₃ for GaOOH/rGO catalyst in different CO₂ partial pressures at 2 V vs. RHE. When the CO₂ partial pressures range from 15% to 100%, the productivity of NaHCO₃ is comparable (from 284.8 to 293.9 mmol L⁻¹ cm⁻² h⁻¹). Therefore, the partial pressure of CO₂ does not have a significant effect on the produced NaHCO₃.

Original Comment 4. *Strengths and Innovation. Reorganize the Introduction to emphasize process innovation over material-centric descriptions. Meanwhile please give a more detailed introduction of the development of GaOOH/rGO catalyst design.*

Response: Thanks for the kind reminding. We have reorganized the Introduction section in Page 5, line 1–Page 6, line 2. Further, we have included a description of the GaOOH/rGO material design as follows:

Firstly, to prevent the occurrence of side reactions of CO₂RR, we selected Ga-based materials based on the volcano plot curve. Secondly, in order to enhance NO₃RR performance and material conductivity, rGO was selected as the substrate for the Ga-based material. Subsequently, through a one-step hydrothermal method, dispersed Ga nanoparticles and GO were synthesized into GaOOH/rGO composite materials, and characterized by SEM, TEM, XPS, XAS and XRD.

Based on the above description, we have made corresponding modifications as follows.

“In this work, we leverage the ‘octet rule’, a fundamental principle in elemental chemistry, to address the above challenging problem. Generally, the reactivity of a chemical process is intricately linked to the number of electrons involved in migration. Other than hydrogen chemistry, such as HER, a vast range of elements in the periodic table adhere to the octet rule, allowing them to accommodate up to eight electrons in valence shells.²⁰ This characteristic could facilitate maximally eight-electron chemical reactions. For instance, in nitrogen chemistry, the nitrate (NO₃⁻) reduction reaction exemplifies this principle.²¹ NO₃⁻ reduction reaction (NO₃RR) affords eight-electron transport by generating the alkalines of NH₃ and OH⁻ (NO₃RR: NO₃⁻ + 6H₂O + 8e⁻ → NH₃ + 9OH⁻; E⁰ = 0.69 V).²² Consequently, by blending *operando* nitrogen chemistry (*i.e.*, *operando* NO₃RR) into electrified process, it can theoretically achieve the maximum CO₂ conversion ratio of NO₃⁻/CO₂ = 1/9 (NO₃⁻ + 8e⁻ + 6H₂O + 9CO₂ → NH₃ + 9HCO₃⁻). Motivated by the hypothesis, we developed an *operando*-electrified system of delivering two environmental pollutants (CO₂ and NO₃⁻) into an electrochemical cell to produce NaHCO₃ (Fig. 1b–c) despite its productivity still unable to rival Solvay

process, owing to the stumbling scaling relationship (the relationship between productivity and production scale) inside the updated system, *i.e.*, parasitic processes of CO₂RR,^{23, 24} HER ($E^0 = 0$ V)²⁵ and carbon-nitrogen (C–N) coupling reaction (CNR; Supplementary Table 2; $E^0 = 0.15 \sim 0.77$ V).^{26, 27}

Indeed, breaking the scaling relationship is very difficult in the literature. In this study, we aim to modify traditional production methodologies and propose an *operando*-electrified production model to enhance the compatibility of the Solvay process. Here, our density function theory (DFT) simulations further suggest a gallium (Ga)-derived catalyst, where its *CO binding energy is situated at a position far from the volcano apex, indicating it is inert to parasitic processes (CO₂RR and CNR). In this *operando*-electrified system with a Ga-derived catalyst, we reveal an *operando*-positive-coupling phenomenon of reversing the negative impact of the scaling relationship into positive synergy: which expedites CO₂ dissolution, leading to high local alkaline concentration and superior NaHCO₃ productivity.” (Page 5, line 1–Page 6, line 2)

“The first requirement for fixation is the favorable occurrence of NO₃RR to produce NH₃ and OH[−], which facilitates CO₂ dissolution and conversion. As evidenced by the Pourbaix diagram (Fig. 1d),²⁸ NO₃RR can take place in the whole pH range from 0 ~ 14 with thermodynamic reaction potentials from 0.69 to −0.31 V (vs. SHE). When the pH of the system is above 9.25, the product is predominantly NH₃. This will increase the dissolution of CO₂ to provide more feedstock for the reaction. Interestingly, the inherently high thermodynamic potential of NO₃RR can inhibit the HER side process (H₂O – H₂ pairs, or HER), which allows for high reaction selectivity.

The second requirement for fixation is a suitable catalyst that prevents CO₂ from participating in the electrochemical reaction of CO₂RR and CNR. Accordingly, a volcano diagram has been built to predict the activities of common metal elements using *CO adsorption energy as descriptors (Fig. 1e).²⁹ According to the Sabatier principle,³⁰ on the right-hand side of the volcano plot are metal elements with weak adsorption ability for *CO intermediates, while on the left-hand side are those with strong adsorption, both of which lead to compromised CO₂RR activities. It is those metal

elements close to the summit of the volcano that display the best CO₂RR activities. Here, we have calculated the *CO adsorption energy of liquid metal gallium (Ga), and found it situated at a distance far from the volcano apex ($E^*_{\text{CO}} = -1.10$ eV and corresponding to the limiting potential of -1.09 V); therefore, it would be an inert candidate for CO₂RR. Nevertheless, as will be discussed later, Ga active sites have still displayed a physical adsorption to CO₂, which can serve to enrich CO₂ on the electrode surface and facilitate the chemical CO₂-to-HCO₃⁻ conversion.” (Page 6, line 4–Page 6, line 22).

We would like to thank the Editor's invitation to revise the manuscript. We are also appreciated of the reviewer #2 for his/her positive recommendation and constructive comments to improve the quality of this work.

After receiving the reviewer's comments, we have spent several intensive weeks in order to answer the concerns raised by the astute reviewer. We believe we have shifted the work to the next level of proof and gained a much deeper understanding of the system. Please see the following point-by-point response to the reviewer #2's comments. Thank you and best wishes,

Kind regards

The authors

Response to Reviewer #2:

Original Comment: *The manuscript authored by Chen et al. reports an operando-electrified system of delivering two environmental pollutants (CO_2 and NO_3^-) into an electrochemical cell to produce NaHCO_3 by employing a liquid metal-derived hybrid catalyst (GaOOH/rGO). The overall system achieves the maximum NaHCO_3 productivity up to $3.69 \text{ mol L}^{-1} \text{ h}^{-1}$. Overall, the manuscript is logically organized, with its mechanistic interpretation largely supported by both experimental and computational evidence. However, some issues are still in existence in this study. I think a further revision is required for the manuscript to improve its quality.*

Original Comment 1. *It is claimed that NO_3^- is converted into NH_3 during the NO_3RR process, which is also a value-added product. It would be better if the authors could further evaluate the faradaic efficiency and yield rate for NH_3 generation.*

Response: Thanks for the kind reminding. Yes, NH_3 is known as an important by-product of our operando-electrified Solvay system. Therefore, we have determined the NH_3 yield and Faradaic efficiency (FE) of each sample (supplementary Fig. 17). Generally, at the current density of 500 mA cm^{-2} , GaOOH/rGO achieves a yield of $1.29 \text{ mmol cm}^{-2} \text{ h}^{-1}$ and a Faradaic efficiency of 55.3%, outperforming those of GaOOH ($0.66 \text{ mmol cm}^{-2} \text{ h}^{-1}$ & 28.2%) and rGO ($0.78 \text{ mmol cm}^{-2} \text{ h}^{-1}$ & 33.5%). The generation of more NH_3 can promote the operando-electrified Solvay process.

“Firstly, NO_3RR can convert NO_3^- into NH_3 as the dominant product with minor amounts of NO_2^- and N_2H_4 (Supplementary Fig. 16–21).” (Page 8, line 13–Page 8, line 14)

“The concentration of NH_3 byproduct was determined by indophenol blue method with modification.⁶ Initially, a portion of electrolyte was extracted from the electrolytic cell and subsequently diluted to the detection range. Next, 2 mL of the diluted electrolyte was mixed with NaOH solution (2 mL, 1 M) containing of 5 wt% salicylic acid and 5 wt% sodium citrate, followed by the addition of 1 mL of 0.05 M NaClO and 0.2 mL of 1.0 wt% sodium nitroferricyanide solution. After two hours under ambient condition, the absorption peak was measured by UV-vis spectrophotometer at the wavelength of

655 nm.” (Supplementary Methods, Determination of nitrate reduction reaction (NO₃RR) product)

Supplementary Fig. 16. UV-vis calibration curve of NH₃. **a**, the UV-vis spectra of standard NH₄Cl solution with different concentrations. **b**, fitted calibration curve used for estimating NH₃ byproduct.

Supplementary Fig. 17. The NH₃ yield rate and Faradaic efficiency for GaOOH/rGO, GaOOH and rGO samples at a current density of 500 mA cm⁻². **a**, NH₃ yield rate. **b**, Faradaic efficiency of NH₃ yield. At a current density of 500 mA cm⁻², GaOOH/rGO achieved a NH₃ yield rate of 1.29 mmol cm⁻² h⁻¹ and Faradaic efficiency of 55.3%, outperforming the comparison samples of GaOOH (0.66 mmol cm⁻² h⁻¹, 28.2%) and rGO (0.78 mmol cm⁻² h⁻¹, 33.5%).

Original Comment 2. The NO_3^- concentration in the electrolyte is 2 M. The reviewer wonders whether the introduced NO_3^- concentration has influence on the concentration of local alkaine and the productivity of NaHCO_3 .

Response: Generally, NO_3^- reduction reaction (NO_3RR) affords eight-electron transport by generating the alkalines of NH_3 and OH^- (NO_3RR : $\text{NO}_3^- + 6\text{H}_2\text{O} + 8\text{e}^- \rightarrow \text{NH}_3 + 9\text{OH}^-$; $E^0 = 0.69 \text{ V}$). Accordingly to Le Chatelier principle, increasing NO_3^- concentrations will expedite the NO_3RR , thereby producing more local alkaline for CO_2 -to- HCO_3^- conversion.

Accordingly, we have conducted additional tests to determine the productivity of NaHCO_3 under different NO_3^- concentrations for GaOOH/rGO (Supplementary Fig. 30). When the NO_3^- concentration is below 2 M, the electrolyte concentration has a significant impact on the yield, with elevated concentrations promoting CO_2 -to- HCO_3^- conversion (86.7 $\text{mmol L}^{-1} \text{ cm}^{-2} \text{ h}^{-1}$ in 0.5 M NaNO_3 ; 134.3 $\text{mmol L}^{-1} \text{ cm}^{-2} \text{ h}^{-1}$ in 1 M NaNO_3). When the NO_3^- concentration exceeds 2 M, the yield of NaHCO_3 is less affected by the concentrations (293.9 $\text{mmol L}^{-1} \text{ cm}^{-2} \text{ h}^{-1}$ in 2 M NaNO_3 ; 302.7 $\text{mmol L}^{-1} \text{ cm}^{-2} \text{ h}^{-1}$ in 3 M NaNO_3 ; 321.7 $\text{mmol L}^{-1} \text{ cm}^{-2} \text{ h}^{-1}$ in 4 M NaNO_3).

“Further, the effects of NO_3^- concentration and CO_2 partial pressure have also been examined, which shows different impacts on the system (Supplementary Fig. 30–31).”

(Page 8, line 23–Page 9, line 1)

Supplementary Fig. 30. The productivity of HCO_3^- for GaOOH/rGO catalyst in different NaNO_3 concentration at 2 V vs. RHE. The amount of the product NaHCO_3 is

closely related to the amount of NO_3^- . That is, when the NO_3^- concentration is below 2 M, a higher NO_3^- concentration leads to an increase in NaHCO_3 productivity (86.7 $\text{mmol L}^{-1} \text{cm}^{-2} \text{h}^{-1}$ in 0.5 M NaNO_3 ; 134.3 $\text{mmol L}^{-1} \text{cm}^{-2} \text{h}^{-1}$ in 1 M NaNO_3). When the NO_3^- concentration is more than 2 M, the increase rate of NaHCO_3 productivity becomes slow (293.9 $\text{mmol L}^{-1} \text{cm}^{-2} \text{h}^{-1}$ in 2 M NaNO_3 ; 302.7 $\text{mmol L}^{-1} \text{cm}^{-2} \text{h}^{-1}$ in 3 M NaNO_3 ; 321.7 $\text{mmol L}^{-1} \text{cm}^{-2} \text{h}^{-1}$ in 4 M NaNO_3).

Original Comment 3. *All electrochemical experiments should be repeated at least three times to ensure the accuracy of results. And the error bars should be added.*

Response: Thank you for your suggestion. Yes, adding error bars to experimental data is necessary for improving data reliability. We repeated the main electrochemical data in the manuscript, plotted error bars in the data graphs, and updated the figures in the manuscript.

Figure 2e. the corresponding NaHCO_3 productivities

Figure 2f. the NaHCO_3 productivities of GaOOH/rGO, GaOOH and rGO catalysts at

different applied potentials.

Figure 2g. comparison experiments for producing NaHCO₃ in 2 M Na₂SO₄ solution with CO₂ pumping and 2 M NaNO₃ solution with Ar pumping, respectively.

Supplementary Fig. 12. a, The NaHCO₃ productivity of GaOOH catalyst in 2 M NaNO₃ aqueous solution with CO₂ pumping.

Supplementary Fig. 12. b, The NaHCO₃ productivity of rGO catalyst in 2 M NaNO₃ aqueous solution with CO₂ pumping.

Supplementary Fig. 30. The productivity of NaHCO₃ for GaOOH/rGO catalyst in different NaNO₃ concentrations at 2 V vs. RHE. The amount of the product NaHCO₃ is closely related to the amount of NO₃⁻. That is, when the NO₃⁻ concentration is below 2 M, a higher NO₃⁻ concentration leads to an increase in NaHCO₃ productivity (86.7 mmol L⁻¹ cm⁻² h⁻¹ in 0.5 M NaNO₃; 134.3 mmol L⁻¹ cm⁻² h⁻¹ in 1 M NaNO₃). When the NO₃⁻ concentration is more than 2 M, the increase rate of NaHCO₃ productivity becomes slow (293.9 mmol L⁻¹ cm⁻² h⁻¹ in 2 M NaNO₃; 302.7 mmol L⁻¹ cm⁻² h⁻¹ in 3 M NaNO₃; 321.7 mmol L⁻¹ cm⁻² h⁻¹ in 4 M NaNO₃).

Supplementary Fig. 31. The productivity of NaHCO₃ for GaOOH/rGO catalyst in different CO₂ partial pressures at 2 V vs. RHE. When the CO₂ partial pressures range from 15% to 100%, the productivity of NaHCO₃ is comparable (from 284.8 to 293.9 mmol L⁻¹ cm⁻² h⁻¹). Therefore, the partial pressure of CO₂ does not have a significant effect on the produced NaHCO₃.

Figure 4b. the NaHCO₃ productivities of GaOOH/rGO at different electrode areas.

Supplementary Fig. 68. The NaHCO₃ productivities of GaOOH/rGO at a current of 2 A in different flow rates of electrolytes.

Figure 4d. the NaHCO₃ productivities of GaOOH/rGO catalysts with the electrode area of 5 × 5 cm² at different currents.

Original Comment 4. *What is pH value for the electrolyte before testing? And how about that during the operando-electrified process?*

Response: Thank you for your helpful suggestion. We have measured the pH changes of electrolyte before and during electrochemical test for GaOOH/rGO electrode (Supplementary Fig. 14). Specifically, the initial pH of the electrolyte was 6.2. With the reaction proceeding, the pH increased rapidly to 8 and then stabilized at around 8.3.

“During the chronopotentiometry test, we have found the initial pH of electrolyte as 6.2, which then rapidly increased to 8.3 after a few minutes. (Supplementary Fig. 14).”

(Page 8, line 6–Page 8, line 7)

Supplementary Fig. 14. The pH values for GaOOH/rGO catalyst at 500 mA cm⁻² in a 10-minute test.

Original Comment 5. *In Figure 2b, it would be better to use the balls of the same color to represent the same atoms.*

Response: Thank you for the kind reminding. We have corrected the typos and errors as follows:

Figure 1b, the electrofixation of CO_2 and NO_3^- in flow cells.

We would like to thank the Editor's invitation to revise the manuscript. We are also appreciated of the reviewer #3 for his/her constructive comments to improve the quality of this work.

After receiving the reviewer's comments, we have spent several intensive weeks in order to answer the concerns raised by the astute reviewer. We believe we have shifted the work to the next level of proof and gained a much deeper understanding of the system. Please see the following point-by-point response to the reviewer #3's comments. Thank you and best wishes,

Kind regards

The authors

Response to Reviewer #3:

Original Comment: *This work aims to replace the traditional, energy- and cost-intensive Solvay process for NaHCO₃ production with an innovative operando-electrified synthesis approach. Inspired by the octet rule, the authors propose integrating eight-electron nitrogen chemistry to enhance local alkaline generation and CO₂ conversion efficiency. The system reportedly achieves a productivity of 3.69 mol L⁻¹ h⁻¹, which is claimed to exceed the benchmark performance of the Solvay process and surpass existing electrosynthetic methods. While the manuscript presents an interesting and potentially impactful concept, several fundamental issues limit its suitability for publication in its current form. Therefore, I suggest that it be considered for publication after further revision.*

First, although the reported productivity is emphasized as a key highlight, the supporting evidence lacks sufficient rigor and persuasiveness. The experimental design and data interpretation do not convincingly establish the claimed enhancement over industrial benchmarks, and the methodology for productivity quantification requires greater clarity and validation. Second, the manuscript suffers from notable issues in logical flow and coherence. Besides, several sections of the manuscript present incomplete or irrelevant data interpretation. More comprehensive mechanistic elucidation and a clearer structure would be necessary to support the high-level claims made. The following are some of the specific reasons.

Original Comment 1. *The formatting of the figures in the main text is inconsistent. For example, some figures are not enclosed or properly framed. Please standardize the figure formatting throughout the manuscript.*

Response: Thanks for the kind reminding. We have rechecked the format of all figures and corrected the typos and errors as follows:

Figure 2e. the corresponding NaHCO₃ productivities

Figure 2f. the NaHCO₃ productivities of GaOOH/rGO, GaOOH and rGO catalysts at different applied potentials.

Figure 2g. comparison experiments for producing NaHCO₃ in 2 M Na₂SO₄ solution with CO₂ pumping and 2 M NaNO₃ solution with Ar pumping, respectively.

Figure 4b, the NaHCO₃ productivities of GaOOH/rGO at different electrode areas.

Figure 4d. the NaHCO₃ productivities of GaOOH/rGO catalysts with the electrode area of 5 × 5 cm² at different currents.

Figure 1b, the electrofixation of CO₂ and NO₃⁻ in flow cells.

Original Comment 2. *Language and grammar issues: Phrases such as "challenge problem," "Other than hydrogen chemistry," and "maximumly eight-electron chemical reactions" are grammatically incorrect or awkwardly phrased. "Despite of" should be corrected to "despite." "Maximumly" should be "maximally"; "owing to" should be owing to. A thorough language revision is necessary to correct grammar and improve readability.*

Response: We are sorry for the typos and errors. We have rechecked these language and grammar issues, and made corresponding corrections as follows:

“In this work, we leverage the ‘octet rule’, a fundamental principle in elemental chemistry, to address the above challenging problem.” (Page 5, line 1–Page 5, line 2)

“Other than hydrogen chemistry, such as HER, a vast range of elements in the periodic table adhere to the octet rule, allowing them to accommodate up to eight electrons in valence shells.” (Page 5, line 3–Page 5, line 5)

“This characteristic could facilitate maximally eight-electron chemical reactions.” (Page 5, line 5–Page 5, line 6)

“Motivated by the hypothesis, we developed an *operando*-electrified system of delivering two environmental pollutants (CO_2 and NO_3^-) into an electrochemical cell to produce NaHCO_3 (Fig. 1b–c) despite its productivity still unable to rival Solvay process, owing to the stumbling scaling relationship (the relationship between productivity and production scale) inside the updated system” (Page 5, line 11–Page 5, line 15)

“Here, we have calculated the $\ast\text{CO}$ adsorption energy of liquid metal gallium (Ga), and found it situated at a distance far from the volcano apex ($E_{\ast\text{CO}} = -1.10$ eV and corresponding to the limiting potential of -1.09 V)” (Page 6, line 17–Page 6, line 19)

“therefore, it would be an inert candidate for CO_2RR . Nevertheless, as will be discussed later, Ga active sites have still displayed a physical adsorption to CO_2 , which can serve to enrich CO_2 on the electrode surface and facilitate the chemical CO_2 -to- HCO_3^- conversion.” (Page 6, line 19–Page 6, line 22)

“On the other hand, CO_2 in gas chamber enters into cathode chamber *via* the gas

diffusion electrodes” (Page 7, line 20–Page 7, line 21)

“The overall productivity shows a nearly linear correlation relationship to electrode areas (productivity = 0.12 * area + 0.15, Fig. 4b), highlighting the advantages of our module cell in contributing to maximizing the use of electrode surfaces.” (Page 12, line 14–Page 12, line 16)

“This value initializes at 1.04 mol L⁻¹ h⁻¹ at the small flow rate of 1.7 mL min⁻¹ owing to insufficient feedstock supply” (Page 12, line 18–Page 12, line 19)

Original Comment 3. *Define terms like "operando nitrogen chemistry" and "scaling relationship" more precisely.*

Response: Thanks for the kind reminding. We have rechecked these terms in the manuscript and added more detailed definition explanations as follows:

“Consequently, by blending *operando* nitrogen chemistry (*i.e.*, *operando* NO₃RR) into the electrified process, it can theoretically achieve the maximum CO₂ conversion ratio of NO₃⁻/CO₂ = 1/9 (NO₃⁻ + 8e + 6H₂O + 9CO₂ → NH₃ + 9HCO₃⁻)” (Page 5, line 9–Page 5, line 11)

“Motivated by the hypothesis, we developed an *operando*-electrified system of delivering two environmental pollutants (CO₂ and NO₃⁻) into an electrochemical cell to produce NaHCO₃ (Fig. 1b–c) despite its productivity still unable to rival Solvay process, owing to the stumbling scaling relationship (the relationship between productivity and production scale) inside the updated system” (Page 5, line 11–Page 5, line 15)

Original Comment 4. *The introduction does not fully meet the norms and logical expectations of scientific writing. The final paragraph of the Introduction should clearly state the research objective, briefly introduce the core strategy or hypothesis, and highlight the innovation and significance without going too deep into technical details. The current version reads more like a mix of Introduction and Results, making it less effective in providing a concise transition into the body of the paper. Consider streamlining the content, correcting grammatical issues, and restructuring the paragraph to improve clarity, coherence, and academic tone. Major issues: too much*

technical detail for an introduction; language and grammar issues; paragraphs jump between multiple ideas with insufficient transitions.

Response: Thanks for your helpful suggestion. We have made corresponding revision as follows:

“Indeed, breaking the scaling relationship is very difficult in the literature. In this study, we aim to modify traditional production methodologies and propose an *operando*-electrified production model to enhance the compatibility of the Solvay process. Here, our density function theory (DFT) simulations further suggest a gallium (Ga)-derived catalyst, where its *CO binding energy is situated at a position far from the volcano apex, indicating it is inert to parasitic processes (CO₂RR and CNR). In this *operando*-electrified system with a Ga-derived catalyst, we reveal an *operando*-positive-coupling phenomenon of reversing the negative impact of the scaling relationship into positive synergy, which expedites CO₂ dissolution, leading to high local alkaline concentration and superior NaHCO₃ productivity.” (Page 5, line 17–Page 6, line 2)

Original Comment 5. *How to prove the strong interaction between GaOOH nanoparticles and rGO.*

Response: We are sorry for not explaining it clearly. We have verified the strong interaction between GaOOH nanoparticles and rGO by both theoretical simulations and experimental characterizations as follows:

1) Theoretically, we have conducted DFT analyses of electronic properties between GaOOH and GaOOH/rGO (Figure 3c). Density of states (DOS) plots reveal that upon composite formation, the bonding orbital DOS is significantly enhanced, accompanied by bandgap narrowing. The reduced bandgap width drives the material conductivity toward metallic behavior. This result is consistent with Bader charge analysis showing that the charge of Ga atoms decreases from 1.81e to 1.76e, and the adjacent O atoms exhibit a charge of -0.95e. The charge transfer at GaOOH/rGO interface indicates the formation of Ga (GaOOH)-O (rGO)-C (rGO) interaction between GaOOH and rGO.

2) Experimentally, we have conducted XPS analyses to quantify elemental

bonding environments inside GaOOH/rGO. As shown in Supplementary Fig. 9, the Ga 2p peak in GaOOH/rGO shift downward by 0.15 eV relative to pure GaOOH, while the O 1s peak (Ga–OH) shifts by 0.2 eV. These peak shifts are attributed to electron transfer from electron-rich rGO to GaOOH, leading to an increase in outer-shell electrons of Ga and O atoms.

“In addition, the peak shift of Ga and O in GaOOH/rGO relative to pure GaOOH is attributed to electron transfer from electron-rich rGO to GaOOH, which indicates the chemical interaction between GaOOH and rGO (Supplementary Fig. 9).” (Page 7, line 12–Page 7, line 15)

“Secondly, the interaction between GaOOH and rGO is evidenced by density functional theory (DFT) simulations showing charge transfer between Ga (GaOOH)–O (rGO)–C (rGO)” (Page 9, line 22–Page 10, line 1)

Figure 3c. density of state (DOS) profile of GaOOH/rGO and GaOOH, where the inset is the optimized structures and Bader charge transfer.

Supplementary Fig. 9. a, The comparison of Ga 2p peaks for GaOOH/rGO and

GaOOH. **b**, The comparison of O 1s peaks for GaOOH/rGO and GaOOH. The Ga 2p peak in GaOOH/rGO has shifted downward by 0.15 eV relative to pure GaOOH, while the O 1s peak (Ga–OH) has shifted by 0.2 eV. These changes are attributed to electron transfer from electron-rich rGO to GaOOH, leading to an increase in outer-shell electrons of Ga and O atoms and corresponding decrease in binding energies.

Original Comment 6. *For the completeness of the article, to attract more attention and to increase the readership, the introduction should cite some of the latest advances in NO₃RR and CO₂RR, such as: Nat Commun. 2024, 15, 3524; Nat Commun, 2025, 16, 731; J. Am. Chem. Soc. 2025, 147, 8871–8880.*

Response: The recommended literature has been cited as follows:

Ref. 21 Liu, W., et al. Efficient ammonia synthesis from the air using tandem non-thermal plasma and electrocatalysis at ambient conditions. *Nat. Commun.* **15**, 3524 (2024).

Ref. 24 Wang, P., et al. Integrated system for electrolyte recovery, product separation, and CO₂ capture in CO₂ reduction. *Nat. Commun.* **16**, 731 (2025).

Ref. 27 Zhao, C., et al. Tailoring activation intermediates of CO₂ initiates C–N coupling for highly selective urea electrosynthesis. *J. Am. Chem. Soc.* **147**, 8871–8880 (2025).

We would like to thank the Editor's invitation to revise the manuscript. We also appreciate the reviewer #4 for his/her constructive comments to improve the quality of this work.

After receiving the reviewer's comments, we have spent several intensive weeks in order to answer the concerns raised by the astute reviewer. We believe we have shifted the work to the next level of proof and gained a much deeper understanding of the system. Please see the following point-by-point response to the reviewer #4's comments. Thank you and best wishes,

Kind regards

The authors

Response to Reviewer #4:

Original Comment: *In this manuscript, the authors present an operando-electrified Solvay process that utilizes the electrochemical NO_3^- reduction reaction to create a locally alkaline micro-environment, facilitating the CO_2 -to- HCO_3^- conversion. While the productivity of NaHCO_3 is impressive, it appears highly challenging for this operando-electrified Solvay process to replace the traditional Solvay process. In my view, although the performance is noteworthy, the conceptual innovation of this study is somewhat limited. Major concerns that arose during the review of this manuscript included:*

Original Comment 1. *(Bi)carbonate deposition represents a persistent challenge in alkaline/neutral CO_2 electrolysis, significantly compromising both system efficiency and operational durability. This issue not only reduces the CO_2 single-pass conversion efficiency but also blocks the gas diffusion pathway in the gas diffusion electrode (GDE), ultimately deteriorating the CO_2 RR performance. In this work, NO_3 RR is employed to generate a highly alkaline micro-environment, facilitating the CO_2 -to- HCO_3^- conversion, while the GDE is utilized to enhance CO_2 mass transport. However, this design raises concerns regarding potential issues analogous to those encountered in alkaline/neutral CO_2 electrolysis. Specifically, will (bi)carbonate deposit on the GDE surface and within its gas channel? If so, could (bi)carbonate precipitate hinder the diffusion of CO_2 and thereby reduce the productivity of NaHCO_3 ? Will this problem affect the performance of the GaOOH/rGO catalyst and the lifespan of the entire system? Potential solutions to mitigate (bi)carbonate deposition should be discussed. Furthermore, although the authors reported a 200-hour stability of this system, it remains unclear whether the system exhibited performance degradation over time. A detailed analysis of performance decline is essential for identifying failure mechanisms, which in turn would inform the design and optimization of this system for practical applications.*

Response to *“Specifically, will (bi)carbonate deposit on the GDE surface and within its gas channel? If so, could (bi)carbonate precipitate hinder the diffusion of CO_2 and*

thereby reduce the productivity of NaHCO₃?”

Thanks for your kind reminding. Different from CO₂ electrolysis, we have observed seldom (bi)carbonate deposition on GDE surface. We have proposed the origin of this phenomenon as follows (Supplementary Fig. 71):

Firstly, in our *operando*-electrified Solvay system, the dynamic pressure regulation and gas flow flushing were employed. This can maintain a continuous CO₂ supply during testing, therefore regulating the internal pressure of the gas chamber in the electrolytic cell. As a consequence, the CO₂ gas is pumped through GDE, delivering to the electrode surface, effectively flushing away NaHCO₃ deposits and preventing blockages in gas channels. Nat. Commun. 15, 2950 (2024); Chem 10, 3067–3087 (2024)

Secondly, the electrolyte was circulated in a single-pass mode. That is, an electrolyte single-pass flow was adopted to prevent excessive accumulation of NaHCO₃ concentration caused by multiple cycles. This has exerted precise control over the electrolyte environment, attenuating the thermodynamic driving force that propels the processes of NaHCO₃ crystallization and deposition.

To verify above hypothesis, we have used a scanning electron microscope (SEM) to characterize the microscopic structure of electrode material after long-term operation (Supplementary Fig. 73). It demonstrated the absence of NaHCO₃ deposition on the electrode surface, where the material morphology remained nearly unaltered.

Based on the above results, we have added/revised the following sentences and figures to updated manuscript and supplementary information:

Supplementary Fig. 71. The proposed solutions to mitigate NaHCO₃ deposition on electrode surface, especially in long-time operation.

We have proposed the origin of this phenomenon as follows:

Firstly, within the operando-electrified Solvay system, dynamic pressure regulation and gas flow flushing techniques were implemented. This configuration is capable of sustaining a continuous supply of CO₂ during testing, thereby regulating the internal pressure of the gas chamber within the electrolytic cell. Consequently, the CO₂ gas is pumped through GDE and delivered to the electrode surface, thereby effectively flushing away NaHCO₃ deposits and averting blockages in gas channels.^{29,30}

Secondly, the electrolyte was circulated in a single-pass mode. It is evident that an electrolyte single-pass flow was adopted in order to prevent excessive accumulation of NaHCO₃ concentration, which would otherwise have been caused by multiple cycles. This has enabled precise modulation of the electrolyte environment, thereby attenuating the thermodynamic driving force responsible for the processes of NaHCO₃ crystallization and deposition.

In order to verify the aforementioned hypothesis, an attempt was made to characterize the microscopic structure of the electrode material after long-term operation by a scanning electron microscope (SEM; Supplementary Fig. 73). This observation is significant in that it demonstrated the absence of NaHCO₃ deposition on the electrode surface, where the material morphology remained nearly unaltered.

Supplementary Fig. 73. a–b, SEM images of GaOOH/rGO electrode after long-duration stability tests. c–e, EDS mapping images of GaOOH/rGO electrode after long-duration stability tests. f, Elements content of GaOOH/rGO electrode after long duration stability tests.

Response to “Furthermore, although the authors reported a 200-hour stability of this system, it remains unclear whether the system exhibited performance degradation over time.”

Following the reviewer’s comment, we have re-examined the system stability over operation duration, *i.e.*, 2 A for 200 hours. (Figure 4g). For the stability test at 2A, owing to the single-pass mode of the system, the product productivity was detected from the flowed electrolyte. We have determined the initial concentration of NaHCO_3 as 0.433 mol L^{-1} by operation for 22 hours. The NaHCO_3 concentration has been measured at approximately every 20 hours, which are 0.425, 0.429, 0.430, 0.426, 0.426, 0.438 and 0.430 mol L^{-1} for 58, 79, 101, 118, 154, 181 and 200 hours, respectively. Obviously, the bicarbonate concentration remained almost constant during the operational condition.

Further, we have examined the performance of GaOOH/rGO electrode before and after the 200-hour stability test (Supplementary Fig. 72). The system was adjusted to a multiple-cycle electrolyte mode, utilizing a 10-minute chronopotentiometry test

method. Before stability test, the productivity of GaOOH/rGO electrode is $1.42 \text{ mol L}^{-1} \text{ h}^{-1}$ at a current of 2 A. After long-term stability test, the productivity of GaOOH/rGO electrode has declined by only 4.8%. Therefore, both our system and catalyst are very stable for practical applications.

Based on above results, we have added/revised the following sentences and figures in updated manuscript and supplementary information:

Figure 4g. the durability test of GaOOH/rGO in the single-pass mode at a current of 2 A, inset with the measured NaHCO_3 concentration during the 200-hour durability test.

Supplementary Fig. 72. The NaHCO_3 productivity of GaOOH/rGO electrode in 10-minute multiple-cycle tests before and after long-duration stability tests.

Original Comment 2. *What are the NO_3RR performances of the GaOOH/rGO, GaOOH, and rGO catalysts? A closed electron balance should be given.*

Response: Following the reviewer's comment, we have conducted additional tests to

determine the NO₃RR yield and Faradaic efficiency (FE) of GaOOH/rGO, GaOOH, and rGO catalysts (Supplementary Fig. 16–21).

At a current density of 500 mA cm⁻², GaOOH/rGO achieved a NH₃ yield rate of 1.29 mmol cm⁻² h⁻¹ and Faradaic efficiency of 55.3%, outperforming the comparison samples of GaOOH (0.66 mmol cm⁻² h⁻¹, 28.2%) and rGO (0.78 mmol cm⁻² h⁻¹, 33.5%). Other by-products of NO₂⁻ and N₂H₄ have also been measured. The NO₂⁻ yield of GaOOH/rGO was 1.49 mmol cm⁻² h⁻¹ (Faradaic efficiency of 16.0%), while GaOOH was 1.74 mmol cm⁻² h⁻¹ (Faradaic efficiency of 18.6%) and rGO was 1.69 mmol cm⁻² h⁻¹ (Faradaic efficiency of 18.2%), respectively. The N₂H₄ yield of GaOOH/rGO was 0.0142 mmol cm⁻² h⁻¹ (Faradaic efficiency of 1.06%), while the GaOOH was 0.0145 mmol cm⁻² h⁻¹ (Faradaic efficiency of 1.09%) and rGO was 0.0144 mmol cm⁻² h⁻¹ (Faradaic efficiency of 1.08%), respectively.

Based on the above results, we have added the following sentences and figures to the revised manuscript and supplementary information:

“Firstly, NO₃RR can convert NO₃⁻ into NH₃ as the dominant product with minor amounts of NO₂⁻ and N₂H₄ (Supplementary Fig. 16–21).” (Page 8, line 13–Page 8, line 14)

“The concentration of NH₃ byproduct was determined by indophenol blue method with modification.⁶ Initially, a portion of electrolyte was extracted from the electrolytic cell and subsequently diluted to the detection range. Next, 2 mL of the diluted electrolyte was mixed with NaOH solution (2 mL, 1 M) containing of 5 wt% salicylic acid and 5 wt% sodium citrate, followed by the addition of 1 mL of 0.05 M NaClO and 0.2 mL of 1.0 wt% sodium nitroferricyanide solution. After two hours under ambient condition, the absorption peak was measured by UV-vis spectrophotometer at the wavelength of 655 nm.

The concentration of NO₂⁻ byproduct was determined by UV-vis spectrophotometer.⁶ Firstly, a color reagent was prepared as follows: 4 g of p-aminobenzenesulfonamide, 0.2 g of N-(1-naphthyl) ethylenediamine dihydrochloride and 10 mL of phosphoric acid were added into 50 mL of deionized water and mixed thoroughly. Subsequently, 5 mL of the diluted electrolyte and 0.1 mL of the color

reagent were mixed for 20 minutes under ambient condition. The absorption spectrum was measured by UV-vis spectrophotometer at the wavelength of 540 nm.

The concentration of N_2H_4 byproduct was determined by Watt and Chrisp method.⁷ The color reagent was prepared by 5.99 g of para-(dimethylamino)benzaldehyde, 30 mL of concentrated hydrochloric acid and 300 mL of ethanol. Subsequently, 5 mL of the diluted electrolyte and 5 mL of color reagent were mixed for 10 minutes under ambient condition. The absorption spectrum was measured by UV-vis spectrophotometer at the wavelength of 457 nm.” (Supplementary Methods, Determination of nitrate reduction reaction (NO_3RR) product)

Supplementary Fig. 16. UV-vis calibration curve of NH_3 . **a**, the UV-vis spectra of standard NH_4Cl solution with different concentrations. **b**, fitted calibration curve used for estimating NH_3 byproduct.

Supplementary Fig. 17. The NH_3 yield rate and Faradaic efficiency for GaOOH/rGO,

GaOOH and rGO samples at a current density of 500 mA cm^{-2} . **a**, NH_3 yield rate. **b**, Faradaic efficiency of NH_3 yield. At a current density of 500 mA cm^{-2} , GaOOH/rGO achieved a NH_3 yield rate of $1.29 \text{ mmol cm}^{-2} \text{ h}^{-1}$ and Faradaic efficiency of 55.3%, outperforming the comparison samples of GaOOH ($0.66 \text{ mmol cm}^{-2} \text{ h}^{-1}$, 28.2%) and rGO ($0.78 \text{ mmol cm}^{-2} \text{ h}^{-1}$, 33.5%).

Supplementary Fig. 18. UV-vis calibration curve of NO_2^- . **a**, the UV-vis spectra of standard NaNO_2 solution with different concentrations. **b**, fitted calibration curve used for estimating NO_2^- byproduct.

Supplementary Fig. 19. The NO_2^- yield rate and Faradaic efficiency for GaOOH/rGO, GaOOH and rGO samples at a current density of 500 mA cm^{-2} . **a**, yield rate. **b**, Faradaic efficiency. The NO_2^- yield of GaOOH/rGO was $1.49 \text{ mmol cm}^{-2} \text{ h}^{-1}$ (Faradaic efficiency of 16.0%), while GaOOH was $1.74 \text{ mmol cm}^{-2} \text{ h}^{-1}$ (Faradaic efficiency of 18.6%) and rGO was $1.69 \text{ mmol cm}^{-2} \text{ h}^{-1}$ (Faradaic efficiency of 18.2%), respectively.

Supplementary Fig. 20. UV-vis calibration curve of N_2H_4 . **a**, the UV-vis spectra of standard N_2H_4 solution with different concentrations. **b**, fitted calibration curve used for estimating N_2H_4 byproduct.

Supplementary Fig. 21. The N_2H_4 yield rate and Faradaic efficiency for GaOOH/rGO, GaOOH and rGO samples at a current density of $500 mA cm^{-2}$. **a**, yield rate. **b**, Faradaic efficiency. The N_2H_4 yield of GaOOH/rGO was $0.0142 mmol cm^{-2} h^{-1}$ (Faradaic efficiency of 1.06%), while the GaOOH was $0.0145 mmol cm^{-2} h^{-1}$ (Faradaic efficiency of 1.09%) and rGO was $0.0144 mmol cm^{-2} h^{-1}$ (Faradaic efficiency of 1.08%), respectively.

Original Comment 3. *The quality of the in-situ Raman spectra in Figure 3d is too poor to draw definitive conclusions about the presence of HCO_3^- . Moreover, the intensity of the Raman signal is strongly influenced by the test parameters and reaction environment (e.g., bubbles). Therefore, directly comparing the Raman signal intensity is unreasonable. It is recommended to introduce a stable internal standard for the Raman test to improve reliability.*

Response to “*The quality of the in-situ Raman spectra in Figure 3d is too poor to draw definitive conclusions about the presence of HCO_3^-* ”

We are appreciated of the reviewer for his/her valuable comments. Yes, we have noted the relatively weak HCO_3^- signal in the *operando* Raman spectra. We have given the explanations as follows:

Generally, Raman spectrum is a scattering signal with an intrinsic weak intensity. When used to study the structural characteristics of a bulk material, obvious signal peaks can be seen. This is due to the structural crystal lattice that contributes to the overall vibration of the bulk material, leading to enhanced scattered signals.

While in this work, the *operando* Raman only probes the signals of *NHO and HCO_3^- during catalytic processes. The scattered Raman signals are mainly focused on bond vibrations of adsorbed species (like *NHO) on catalyst surfaces, which are known to be very weak as comparison to structural crystal lattices in bulk materials. Consequently, the *operando* Raman signals for catalytic reactions are mostly very weak in the literature (like ORR, NRR and CRR). J. Am. Chem. Soc. 142, 715-719 (2020); Angew. Chem. Int. Edit. 60, 20331-20341 (2021); ACS Nano 14, 11363-11372 (2020) Our *operando* Raman signals in Figure 3d are comparable to the above literature.

To further validate the presence of HCO_3^- and the reliability of the *operando* Raman data, we have conducted additional experiments of *operando* FTIR spectra (Supplementary Fig. 40). The FTIR vibration peak of HCO_3^- (1656 cm^{-1} and 1697 cm^{-1}) is clearly observed, with signal intensity increasing monotonically with applied potential, which is consistent with the *operando* Raman results.

Response to “*It is recommended to introduce a stable internal standard for the Raman*

test to improve reliability”

In the *operando* Raman spectra, a sharp and intense Raman signal at 3000–3600 cm^{-1} , assigned to O–H stretching band of H_2O , was identified. According to previous research, *Chemical Geology* 3–4 (283), 274–278 (2011); *Geoscience Frontiers* 2 (12), 1018–1030 (2021) this can give a stable condition during test. Therefore, we consider that O–H stretching signal peak can serve as an internal standard for calibrating the relative intensities of other product signal peaks. We have conducted the *operando* Raman test four times repeatedly, calibrating the relative intensities, and added error bars to the *operando* Raman results as follows:

Figure 3d, the *operando* Raman spectra of GaOOH/rGO under different applied potentials.

Figure 3e. the relative peak intensity analyses in *operando* Raman test, standardized with the peak intensity of H_2O .

Supplementary Fig. 39. a–b, repetitive *operando* Raman tests for GaOOH/rGO system at a potential range of $-0.5 \text{ V} \sim -1.0 \text{ V}$ vs. RHE.

Supplementary Fig. 40. The *operando* FTIR test of GaOOH/rGO system at a potential range of $-0.6\text{ V} \sim -1.1\text{ V}$ vs. RHE.

Original Comment 4. *In line 168, the authors highlight the charge transfer between Ga and C based on DFT calculations. Can this conclusion be further supported experimentally by XPS and XAS data? Comparing the XPS and XAS spectra of GaOOH and GaOOH/rGO catalysts would help clarify the direction of electron transfer and confirm the interaction between GaOOH and rGO.*

Response: Thanks for the kind reminding. Firstly, we have conducted XPS analyses for GaOOH and GaOOH/rGO catalysts (Supplementary Fig. 9). The Ga 2p peak in GaOOH/rGO has shifted downward by 0.15 eV relative to pure GaOOH, while the O 1s peak (Ga–OH) shifted by 0.2 eV. These peak shifts are attributed to electron transfer from electron-rich rGO to GaOOH, leading to an increase in outer-shell electrons of Ga and O atoms.

Further, owing to the limitation of XAS resources, we have only conducted the XAS of GaOOH/rGO. In pure GaOOH, the valence state of Ga is +3, which is the same as Ga₂O₃. So we compared GaOOH/rGO with the standard sample Ga₂O₃. In XANES spectra, a shift of the Ga absorption edge toward lower energy was observed in GaOOH/rGO, indicating that the valence state of Ga is lower than +3 (compared to Ga₂O₃). This finding is in line with the results of XPS, which demonstrate the

interaction between GaOOH and rGO.

“In addition, the peak shift of Ga and O in GaOOH/rGO relative to pure GaOOH is attributed to electron transfer from electron-rich rGO to GaOOH, which indicates the chemical interaction between GaOOH and rGO (Supplementary Fig. 9).” (Page 7, line 12–Page 7, line 15)

Supplementary Fig. 9. a, The comparison of Ga 2p peaks for GaOOH/rGO and GaOOH. **b**, The comparison of O 1s peaks for GaOOH/rGO and GaOOH. The Ga 2p peak in GaOOH/rGO has shifted downward by 0.15 eV relative to pure GaOOH, while the O 1s peak (Ga–OH) has shifted by 0.2 eV. These changes are attributed to electron transfer from electron-rich rGO to GaOOH, leading to an increase in outer-shell electrons of Ga and O atoms and corresponding decrease in binding energies.

Figure 3a. X-ray absorption near edge structure (XANES) spectra of GaOOH/rGO, Ga₂O₃ and Ga foils

Original Comment 5. *Detailed characterization of the GaOOH/rGO and GaOOH catalysts, both during and after NO₃RR, is necessary to determine the true structure of catalysts under reduction potential. The applied reduction potential ranges from $-0.7\text{ V} \sim -2\text{ V}$ vs RHE, it is essential to address whether GaOOH will be reduced to metallic Ga under these conditions. The DFT calculation model should align with the actual structure of the catalysts under operating conditions, rather than the pristine GaOOH structure.*

Response: Thanks for your kind reminding. We have conducted a series of characterizations for GaOOH/rGO before and after test, and given the following response:

1) Firstly, we have characterized the electrodes by Raman spectra (Supplementary Fig. 41). GaOOH/rGO only exhibited the D and G bands from carbon species, which is similar to the case of rGO, and there was no significant change in Raman signals before and after electrochemical testing. While for the GaOOH sample, several Raman vibration signals can be observed that belong to Ga–O (272, 429, 515, 605 and 694 cm^{-1}), with seldom change before and after testing.

2) Secondly, we have conducted FTIR characterizations (Supplementary Fig. 42). The GaOOH/rGO has exhibited vibration signals of Ga–O (580 cm^{-1}), Ga–OH (945 and 1065 cm^{-1}), O–H (2841 and 2918 cm^{-1}), and C=C (1558 cm^{-1}), which are similar to GaOOH and different from rGO. Notably, there are seldom changes in peak intensities observed before and after test.

3) Thirdly, XPS was performed before and after electrochemical test (Figure 2c; Supplementary Fig. 6–8; Supplementary Fig. 43–44). The Ga 2p characteristic peak of GaOOH/rGO samples has exhibited characteristics of trivalent Ga before and after test (in Ga-metal, the binding energies for 2p_{3/2} and 2p_{1/2} are 1116.5 and 1143.3 eV, respectively). ^{Superlattices and Microstruct. 120, 90-100 (2018)}. In the O 1s spectra, the peaks for Ga–O, Ga–OH, C=O, C–O–C, and Na KLL (derived from electrolyte) were detected. In comparison to the characteristic peaks before test, the binding energies of all subsequent peaks seldom change. In the C 1s spectra, C=C, C–C, C–O–C, O–C=O, and π – π^* peaks

were detected. There were seldom changes in the positions of these peaks.

Based on the above experimental data of Raman, FTIR and XPS, we conclude that GaOOH/rGO is a stable metal-oxyhydroxide hybrid material. One of the reasons is the *operando* generation of an alkaline environment during the electrocatalytic reduction reaction contributing to the stability of oxyhydroxide. ^{Langmuir 34, 7604-7611 (2018)}. According to previous literature, metal oxyhydroxide can serve as an active material to catalyze the reduction reaction. ^{ACS Nano 16, 8213-8222 (2022); Adv. Energy Mater. 12, 2200077 (2022); Nat. Commun. 14, 2040 (2023)}. Secondly, the strong interaction between rGO and GaOOH also stabilizes the active valence state of Ga, thereby inhibiting excessive reduction. That is, the oxygen-containing functional groups on the rGO surface form a coordination bond (Ga-O-C) with Ga³⁺, constraining and preventing excessive electron transfer that leads to the formation of Ga⁰. ^{Nat. Commun. 9, 935 (2018); ACS Sustain. Chem. Eng. 5, 3186-3194 (2017)}. Thirdly, the high conductivity of rGO promotes the directed transfer of electrons to active sites, thus avoiding the formation of elemental Ga due to excess electrons. ^{J. Phys. Chem. C 119, 7069-7075 (2015)}. As a consequence, despite Ga requiring electrons to initiate the reduction reaction (*e.g.*, the reduction of the target substrate NO₃⁻), the high conductivity of rGO can efficiently transfer electrons to the Ga sites, thereby inhibiting over-reduction of Ga³⁺ to Ga⁰.

Supplementary Fig. 41. The Raman spectra for electrodes before and after the electrochemical testing. **a**, GaOOH/rGO, **b**, GaOOH and **c**, rGO, respectively. In Raman spectra, GaOOH/rGO only exhibited the D and G bands from carbon species, which is similar to the case of rGO, and there was no significant change in Raman signals before and after electrochemical testing. While for the GaOOH sample, several Raman vibration signals can be observed that belong to Ga–O (272, 429, 515, 605 and 694 cm⁻¹), with seldom change before and after tests.

Supplementary Fig. 42. The FTIR spectra for electrodes before and after the electrochemical testing. **a**, GaOOH/rGO, **b**, GaOOH and **c**, rGO, respectively. The GaOOH/rGO has exhibited vibration signals of Ga–O (580 cm^{-1}), Ga–OH (945 and 1065 cm^{-1}), O–H (2841 and 2918 cm^{-1}), and C=C (1558 cm^{-1}), which are similar to GaOOH and different from rGO. Notably, there are seldom changes in peak intensities observed before and after test.

Supplementary Fig. 43. The XPS spectra for GaOOH/rGO catalyst after electrochemical testing. **a**, XPS survey of GaOOH/rGO. **b**, XPS Ga 2p peaks of GaOOH/rGO. **c**, XPS O 1s peaks of GaOOH/rGO. **d**, XPS C 1s peaks of GaOOH/rGO. The Ga 2p characteristic peak for the GaOOH/rGO catalyst exhibited a slight shift towards lower binding energy, consistent with the characteristics of trivalent Ga (in Ga-metal, the binding energies for $2p_{3/2}$ and $2p_{1/2}$ are 1116.5 and 1143.3 eV, respectively).²¹ In the O 1s spectra, those peaks for Ga–O, Ga–OH, C=O, C–O–C, and Na KLL (derived from electrolyte) can be detected. As comparison to the characteristic peaks observed before testing, the binding energies of all peaks have exhibited seldom changes.

Supplementary Fig. 44. The XPS spectra for GaOOH catalyst after electrochemical testing. **a**, XPS survey of GaOOH. **b**, XPS Ga 2p peaks of GaOOH. **c**, XPS O 1s peaks of GaOOH. These characteristic signal shows seldom change.

“Based on above data, the material stability of GaOOH/rGO catalyst has been discussed.

Firstly, the electrodes were characterized by Raman spectra (Supplementary Fig. 41). GaOOH/rGO exhibited the D and G bands from carbon species, which is similar to the case of rGO, and there was no significant change in Raman signals before and after electrochemical testing. In the case of the GaOOH sample, several Raman vibration signals belonging to Ga–O (272, 429, 515, 605 and 694 cm^{-1}) were observed, with no significant alterations noted before and after test.

Secondly, FTIR characterizations were conducted (Supplementary Fig. 42). The vibration signals of Ga–O (580 cm^{-1}), Ga–OH (945 and 1065 cm^{-1}), O–H (2841 and 2918 cm^{-1}), and C=C (1558 cm^{-1}) have been exhibited by the GaOOH/rGO, and these are similar to those of GaOOH and different from those of rGO. It is noteworthy that

there is frequently an absence of alterations in peak intensity levels.

Thirdly, XPS was performed before and after electrochemical testing (Figure 2c, Supplementary Fig. 6–8; Supplementary Fig. 43–44). The Ga 2p characteristic peak of the GaOOH/rGO samples has exhibited characteristics of trivalent Ga before and after testing (in Ga-metal, the binding energies for 2p_{3/2} and 2p_{1/2} are 1116.5 and 1143.3 eV, respectively).²¹ In the O 1s spectra, the peaks for Ga–O, Ga–OH, C=O, C–O–C, and Na KLL (derived from electrolyte) were detected. As comparison to the characteristic peaks before the test, the binding energies of all peaks rarely undergo alteration. The positions of these peaks remained relatively static.

It is evident from the experimental data pertaining to Raman, FTIR and XPS that GaOOH/rGO is a stable metal-oxyhydroxide hybrid material. One of the factors contributing to this phenomenon is the *operando* generation of an alkaline environment during the electrocatalytic reduction reaction, which has been shown to contribute to the stability of oxyhydroxide.²² As stated in previous literature, metal oxyhydroxide has been identified as an active material with the capacity to catalyze the reduction reaction.^{23–25} Secondly, the strong interaction between rGO and GaOOH also stabilizes the active valence state of Ga, thereby inhibiting excessive reduction. It can thus be concluded that the oxygen-containing functional groups on the rGO surface form a coordination bond (Ga–O–C) with Ga³⁺, thereby constraining and preventing excessive electron transfer that leads to the formation of Ga⁰.^{26, 27} Thirdly, the high conductivity of rGO promotes the directed transfer of electrons to active sites, thus avoiding the formation of elemental Ga due to excess electrons.²⁸ Consequently, despite Ga requiring electrons to initiate the reduction reaction (*e.g.*, the reduction of the target substrate NO₃[−]), the high conductivity of rGO can efficiently transfer electrons to the Ga sites, thereby inhibiting over-reduction of Ga³⁺ to Ga⁰.” (discussion following Supplementary Fig. 44)

Original Comment 6. *The discussion of the DFT calculations regarding the difference charge density (line 202) is confusing. First, the active site of GaOOH/rGO must be explicitly defined. Secondly, if electron-rich CO₂ contributes electrons to GaOOH/rGO, will this lead to an increased electron density of GaOOH/rGO, thereby hindering the adsorption of negatively charged NO₃⁻? The authors claim that physically adsorbed CO₂ promotes the NO₃RR at the GaOOH/rGO catalyst based on DFT simulations. Can the same enhancement in selectivity or activity for NO₃RR be experimentally observed on the GaOOH/rGO catalyst?*

Response: We are sorry for not explaining it clearly. Our response is as follows:

1) The active site for the reaction is Ga of GaOOH/rGO. We have labelled these sites in the DFT theoretical models (Supplementary Fig. 46–47).

2) The adsorption of CO₂ on catalyst surface has not hindered the adsorption of negatively charged NO₃⁻. The reason is proposed as follows: *i*) the adsorption of CO₂ on the catalyst surface is not static. Rather, CO₂ is continuously consumed *via* CO₂-to-HCO₃⁻ conversion, indicating its influence on the charge distribution is short-lived. This is consistent with BET and *operando* FTIR spectra, showing that CO₂ on the catalyst surface is primarily physically adsorbed (Supplementary Fig. 38; Supplementary Fig. 40), with only a small amount being chemically adsorbed. *ii*) In DFT theoretical models, the distance between CO₂ and Ga active site is 4.00 Å, which exceeds chemical bond length. Therefore, the influence of such weak interaction is limited. As a result, the difference charge density and density of states plots (Supplementary Fig. 48–67) show the presence of CO₂ only introducing slight increase in electron distribution. The Gibbs free energy only changes by 0.08 eV between the presence and absence of CO₂ (Figure 3f). The change in the electron distribution caused by CO₂ adsorption is insufficient to hinder the adsorption of NO₃⁻.

3) Experimentally, we have evaluated the performance of NO₃RR in the presence and absence of CO₂ (Supplementary Fig. 17; Supplementary Fig. 22). At the current density of 500 mA cm⁻² under Ar atmosphere, GaOOH/rGO shows good NO₃RR activity with a NH₃ yield rate of 0.96 mmol cm⁻² h⁻¹ and Faradic efficiency of 41.2%.

In great contrast, at the current density of 500 mA cm^{-2} under CO_2 atmosphere, GaOOH/rGO shows enhanced NO_3RR activity with a NH_3 yield rate of $1.29 \text{ mmol cm}^{-2} \text{ h}^{-1}$ and Faradic efficiency of 55.3%. Therefore, we conclude the enhancement effect of CO_2 also for NO_3RR , which will generate more alkaline (NH_3 and OH^-) for *operando*-electrified Solvay system.

Based on the above results, the following sentences and figures have been added to the updated manuscript:

“As comparison to Ar, pumping CO_2 into the system can accelerate NO_3^- -to- NH_3 conversion (Supplementary Fig. 22).” (Page 8, line 14–Page 8, line 15)

“Interestingly, most adsorption energies of intermediates show a slight drop in the presence of CO_2 , owing to the electron-rich CO_2 that contributes a brief electrostatic induction effect to GaOOH/rGO, leading to a decreased electron density at the active sites for adsorbing intermediates.” (Page 11, line 18–Page 11, line 21)

Supplementary Fig. 22. The NH_3 yield rate and Faradaic efficiency for GaOOH/rGO at a current density of 500 mA cm^{-2} , with Ar and CO_2 pumping, respectively. **a**, yield rate. **b**, Faradaic efficiency.

Supplementary Fig. 38. The BET test of GaOOH/rGO in the CO₂ atmosphere. The BET results indicate the GaOOH/rGO with a pore area of 62.2 m² g⁻¹ and a CO₂ adsorption capacity of 7.98 cm³ g⁻¹. This result indicates GaOOH/rGO exhibiting good CO₂ adsorption capabilities.

Supplementary Fig. 40. The *operando* FTIR test of GaOOH/rGO catalyst at a potential range of -0.6 V ~ -1.1 V vs. RHE.

Supplementary Fig. 46. The optimized structures of reaction intermediates for NO₃RR on GaOOH/rGO surface, where the adsorbates from **a–j** are *NO₃⁻, *NO₃H, *NO₂, *NO₂H, *NO, *NHO, *NHOH, *NH, *NH₂ and *NH₃, respectively.

Supplementary Fig. 47. The optimized structures of reaction intermediates for NO₃RR on GaOOH/rGO surface with CO₂, the adsorbates from **a–j** are *NO₃⁻, *NO₃H, *NO₂, *NO₂H, *NO, *NHO, *NHOH, *NH, *NH₂ and *NH₃, respectively.

Original Comment 7. *Are there any by-products, such as Na₂CO₃, generated during the process?*

Response: Thanks for your kind reminding. We have conducted XRD to determine the possible Na₂CO₃ byproduct (Supplementary Fig. 70).

Specifically, we have performed XRD analyses on commercial Na₂CO₃ and NaHCO₃, and then compared them with our synthesized product. The results show that the XRD pattern of our synthesized product is in excellent agreement with the commercial NaHCO₃ standard, while displaying distinct differences from Na₂CO₃. Based on the above results, the following sentences and figures have been added to the updated manuscript:

“For example, the as-produced product has been collected, concentrated, and separated

from the electrolytes according to solubility, which can achieve up to 1.30 kg with a purity comparable to that of commercial NaHCO_3 (Fig. 4f; Supplementary Fig. 69–70; Supplementary Table 7).” (Page 13, line 11–Page 13, line 13)

Supplementary Fig. 70. The XRD patterns of as-synthesized NaHCO_3 , commercial NaHCO_3 and commercial Na_2CO_3 .

Original Comment 8. *While this work highlights the advantages of producing NaHCO_3 in productivity, industrialization remains constrained by other factors, such as system stability. Could the authors comment on the industrialization prospects of this operando-electrified Solvay process?*

Response: Thank you for useful suggestion. We have discussed the potential industrialization of *operando*-electrified Solvay process, and given responses as follows:

1) Indeed, the technological breakthrough in system stability is an essential condition. In our *operando*-electrified Solvay system, we have attempted to achieve catalyst and cell design optimization. The optimized GaOOH/rGO catalyst exhibits only 4.8% decay in NaHCO_3 yield after 200 hours of continuous operation at 2A, attributed to its surface pH-swing arising from NO_3RR activity and CO_2RR passivity. Further, the flow-cell configuration suppresses NaHCO_3 deposition *via operando* CO_2 flushing. The good system stability is confirmed by post-operation SEM images showing little electrode morphology change (Supplementary Fig. 73).

2) Next, the production cost is another vital criterion for industrialization. In the

traditional Solvay process, the ammonia and alkaline as feedstock constitute a substantial proportion of the overall cost. ^{Environ. Chem. 21, 1–14 (2024)}. In this respect, our *operando*-electrified Solvay process represents a significant reduction in the use of expensive performed raw materials. Our cost analysis reveals that the *operando*-electrified Solvay system shows significantly lower cost than traditional Solvay process.

3) For promoting further industrialization, our *operando*-electrified Solvay system still needs further improvement. Firstly, the issue of electricity costs must be addressed. The current global distribution of electricity is uneven, and the market for clean electricity has yet to be fully established. It is anticipated that a reduction in electricity costs will result in a period of significant development for our electrified plan. Secondly, the present system is deficient in the realm of large-scale pilot testing and verification. The processes of industrialization and expansion of production are not linear; rather, they are complex and multifaceted, and the engineering applications and process design involved require further exploration.

Based on the above results, the following sentences and figures have been added to the updated supplementary information:

“Based on above data, the potential industrialization of *operando*-electrified Solvay process has been discussed.

1) Firstly, the technological breakthrough in system stability is an essential condition. In our *operando*-electrified Solvay system, we have attempted to achieve catalyst and cell design optimization. The optimized GaOOH/rGO catalyst exhibits only ~ 5% decay in NaHCO₃ yield after 200 hours of continuous operation at 2A, attributed to its surface pH-swing arising from NO₃RR activity and CO₂RR passivity. Furthermore, the flow-cell configuration suppresses NaHCO₃ deposition *via operando* CO₂ flushing. The good system stability is confirmed by post-operation SEM images showing little electrode morphology change (Supplementary Fig. 73).

2) Next, the production cost is another vital criterion for industrialization. In the traditional Solvay process, the ammonia and alkaline as feedstock constitute a substantial proportion of the overall cost.³¹. In this respect, our *operando*-electrified Solvay process represents a significant reduction in the use of expensive performed raw

materials. Our cost analysis reveals that the *operando*-electrified Solvay system has a significantly lower cost than the traditional Solvay process.

3) For promoting further industrialization, our *operando*-electrified Solvay system still needs further improvement. Firstly, the issue of electricity costs must be addressed. The current global distribution of electricity is uneven, and the market for clean electricity has yet to be fully established. It is anticipated that a reduction in electricity costs will result in a period of significant development for our electrified plan. Secondly, the present system is deficient in the realm of large-scale pilot testing and verification. The processes of industrialization and expansion of production are not linear; rather, they are complex and multifaceted, and the engineering applications and process design involved require further exploration.” (discussion following Supplementary Fig. 74)

We would like to thank the Editor's invitation to revise the manuscript. We are also appreciated of the reviewer #5 for his/her constructive comments to improve the quality of this work.

After receiving the reviewer's comments, we have spent several intensive weeks in order to answer the concerns raised by the astute reviewer. We believe we have shifted the work to the next level of proof and gained a much deeper understanding of the system. Please see the following point-by-point response to the reviewer #5's comments. Thank you and best wishes,

Kind regards

The authors

Response to Reviewer #5:

Original Comment: *In this work, the authors present an interesting method to produce NaHCO₃ with the assistance of electrocatalytic nitrate reduction. The productivity of HCO₃⁻ is much higher than the traditional Solvay process. However, I feel there are some issues that the authors need to carefully address. My detailed comments are as follows.*

Original Comment 1. *The authors claimed that there is a “strong synergistic effect between GaOOH and rGO”. However, the electrocatalytic performance of GaOOH, rGO, and GaOOH/rGO do not show much difference (Fig.2e, Supplementary Fig.9-10). It seems that the synergistic effect between GaOOH and rGO is weak.*

Response: We are sorry for not explaining it clearly. We have re-examined the data for the electrocatalytic performances of GaOOH, rGO, and GaOOH/rGO. Indeed, as pointed out by the reviewer, Supplementary Fig. 9 of original manuscript (Supplementary Fig. 12 in revised manuscript) shows the NaHCO₃ productivity of GaOOH and rGO comparable to GaOOH/rGO at different current densities, for example, 65.8, 71.0 vs. 87.5 mmol L⁻¹ h⁻¹ cm⁻² at 100 mA cm⁻², 202.5, 213.7 vs. 251.9 mmol L⁻¹ h⁻¹ cm⁻² at 300 mA cm⁻² and 343.1, 355.8 vs. 374.4 mmol L⁻¹ h⁻¹ cm⁻² at 500 mA cm⁻², respectively.

While we can also see the different average potentials to achieve these current densities by chronopotentiometry tests in Supplementary Fig. 10 of the original manuscript (Supplementary Fig. 13 in revised manuscript), where GaOOH/rGO requires the lowest potentials as comparable to GaOOH and rGO, for example, -0.72 V vs. -0.98 V and -0.91 V for 100 mA cm⁻², -1.42 V vs. -1.59 V and -1.59 V for 300 mA cm⁻² and -2.00 V vs. -2.08 V and -2.16 V for 500 mA cm⁻², respectively. This result indicates that GaOOH/rGO is more energy efficient than GaOOH and rGO counterparts.

To further verify the excellent performance of GaOOH/rGO, we have conducted chronoamperometry experiments (Figure 2f). The results demonstrate that GaOOH/rGO outperforms the comparison samples (GaOOH and rGO) at different

applied potentials, for example, 162.2 vs. 80.4 and 67.4 mmol L⁻¹ h⁻¹ cm⁻² at 1 V, 243.6 vs. 141.7 and 126.7 mmol L⁻¹ h⁻¹ cm⁻² at 1.5 V, 293.0 vs. 228.3 and 218.8 mmol L⁻¹ h⁻¹ cm⁻² at 2 V, respectively.

The enhanced performances can be attributed to the interaction between GaOOH and rGO, which has been examined by both theoretical and experimental studies. Theoretically, we have conducted DFT analyses of electronic properties between GaOOH and GaOOH/rGO (Figure 3c). Density of states (DOS) plots reveal that upon composite formation, the bonding orbital DOS is significantly enhanced, accompanied by bandgap narrowing. The reduced bandgap width drives the material conductivity toward metallic behavior. This result is consistent with Bader charge analysis, showing that the charge of Ga atoms decreases from 1.81e to 1.76e, and the adjacent O atoms exhibit a charge of -0.95e. The charge transfer at GaOOH/rGO interface indicates the formation of Ga (GaOOH)-O (rGO)-C (rGO) interaction between GaOOH and rGO. Experimentally, we have conducted XPS analyses to quantify elemental bonding environments inside GaOOH/rGO. As shown in Supplementary Fig. 9, the Ga 2p peak in GaOOH/rGO shifts downward by 0.15 eV relative to pure GaOOH, while the O 1s peak (Ga-OH) shifts by 0.2 eV. These peak shifts are attributed to electron transfer from electron-rich rGO to GaOOH, leading to an increase in outer-shell electrons of Ga and O atoms.

Finally, to improve the scientific content of this work, we have removed the term “strong” in the statement (Page 9, line 22).

Based on the above results, we have revised/added the following sentences and figures in the updated manuscript:

“In addition, the peak shift of Ga and O in GaOOH/rGO relative to pure GaOOH is attributed to electron transfer from electron-rich rGO to GaOOH, which indicates the chemical interaction between GaOOH and rGO (Supplementary Fig. 9).” (Page 7, line 12–Page 7, line 14)

“Secondly, the interaction between GaOOH and rGO is evidenced by density functional theory (DFT) simulations showing charge transfer between Ga (GaOOH)-O (rGO)-C (rGO)” (Page 9, line 22–Page 10, line 1)

Figure 2f. the NaHCO₃ productivities of GaOOH/rGO, GaOOH and rGO catalysts at different applied potentials.

Figure 3c. density of state (DOS) profile of GaOOH/rGO and GaOOH, where the inset is the optimized structures and Bader charge transfer.

Supplementary Fig. 9. a, The comparison of Ga 2p peaks for GaOOH/rGO and GaOOH. **b,** The comparison of O 1s peaks for GaOOH/rGO and GaOOH. The Ga 2p

peak in GaOOH/rGO has shifted downward by 0.15 eV relative to pure GaOOH, while the O 1s peak (Ga-OH) has shifted by 0.2 eV. These changes are attributed to electron transfer from electron-rich rGO to GaOOH, leading to an increase in outer-shell electrons of Ga and O atoms and corresponding decrease in binding energies.

Original Comment 2. *How about the purity of the generated NaHCO₃? The product purity is closely related to the purification cost, which should also be taken into account when calculating the economic costs. Btw, I suggest the authors to provide the details for the product collection and purification.*

Response: Thanks for your kind suggestion. We have given the response as follows:

1) We have determined the purity of NaHCO₃ by XRD patterns (Supplementary Fig. 70). We have tested XRD on commercial Na₂CO₃ and NaHCO₃, and then compared them with our synthesized product. The results show that the XRD pattern of our synthesized product is in excellent agreement with the commercial NaHCO₃ standard, while displaying its high purity.

2) The product collection and purification procedure is described as follows: it is known that the reaction electrolyte contains a complex mixture of NaNO₃, NH₄NO₃, Na₂CO₃, NaHCO₃, NH₄HCO₃ and (NH₄)₂CO₃. These chemicals have shown different physical and chemical properties, particularly different solubilities in pH environments (Supplementary Table 7). Therefore, we adjust the pH of the electrolyte, and facilitate the purification of NaHCO₃. Experimentally, we have adjusted the pH of the electrolyte to approximately 8.3, at which point the solution contains only Na⁺, NH₄⁺, HCO₃⁻, and NO₃⁻. The aforementioned solutions (*e.g.*, 20 mL) were then evaporated at 35°C until the onset of crystallization (~ 5 hours). Next, the solution was immediately frozen for over 2 hours, where white crystal precipitate formed as the final product NaHCO₃.

Based on the above results, we have revised/added the following sentences and figures in the updated manuscript:

“For example, the as-produced product has been collected, concentrated, and separated from the electrolytes according to solubility, which can achieve up to 1.30 kg with a purity comparable to that of commercial NaHCO₃ (Fig. 4f; Supplementary Fig. 69–70;

Supplementary Table 7).” (Page 13, line 11–Page 13, line 13)

“After electrolysis, the electrolyte (20mL) contains a mixture of NaNO_3 , NH_4NO_3 , Na_2CO_3 , NaHCO_3 , NH_4HCO_3 and $(\text{NH}_4)_2\text{CO}_3$. These chemicals have shown different physical/chemical properties, particularly different solubilities in discrepant pH environments (Supplementary Table 7). Therefore, we have adjusted the pH of the electrolyte, and facilitate the purification of NaHCO_3 . Experimentally, we have adjusted the pH of the electrolyte to ~ 8.3 by sulfuric acid. The as-formed solution was then evaporated in a 35°C until the onset of crystallization (~ 5 hours). Immediately, the solution was frozen for 2 hours, where the as-precipitated white crystal is the target product of NaHCO_3 .” (Supplementary Methods, Product purification and collection)

Supplementary Fig. 70. The XRD patterns of as-synthesized NaHCO_3 , commercial NaHCO_3 and commercial Na_2CO_3 .

Supplementary Table 7. The solubility of NaNO_3 , NH_4HCO_3 , NH_4NO_3 and NaHCO_3 .¹¹⁶

solute	solubility g / 100 g H_2O (20°C)
NaNO_3	87.2
NH_4HCO_3	21.7
NH_4NO_3	101.7
NaHCO_3	9.6

Original Comment 3. *During the CO₂-assisted electrocatalytic NO₃⁻ reduction, NH₄⁺ will be produced in the aqueous solution. This means that, NaHCO₃ and NH₄HCO₃ are produced simultaneously. For the calculation of HCO₃⁻ productivity, the authors measure the total concentration of HCO₃⁻ in the solution, which comes from both NaHCO₃ and NH₄HCO₃, but only NaHCO₃ is the target product. So, I think calculating NaHCO₃ productivity is more reasonable. Similarly, the authors should compare the productivity of NaHCO₃ in CO₂-purged NaNO₃ and Na₂SO₄ solution, to prove that NO₃⁻ reduction indeed accelerate the generation of NaHCO₃.*

Response: Thanks for the kind reminding, we have conducted calculations and experiments, and given the response as follows:

1) Yes, as pointed out by the reviewer, there are also NH₄⁺ in the electrolyte after reaction, originating from NO₃⁻-to-NH₄⁺ conversion. With the elongation of reaction duration, the ratio of Na⁺/NH₄⁺ concentration changes continuously. Because of it difficulty in determining the ratio of NaHCO₃/NH₄HCO₃, we have conducted calculations for the concentration change during the reaction process (Supplementary Fig. 69).

The system starts by introducing NaNO₃ electrolyte containing NO₃⁻ (2 mol L⁻¹) and Na⁺ (2 mol L⁻¹). With the polarization at 2 A from 0 ~ 15 hours, the concentrations of NO₃⁻ decrease linearly from 2 mol L⁻¹ to 0.6 mol L⁻¹, while that of NH₄⁺ increases linearly from 0 mol L⁻¹ to 1.4 mol L⁻¹, which is in line with the NO₃⁻-to-NH₄⁺ conversion.

In great contrast, the concentration of Na⁺ remains stable at 2 mol L⁻¹ in the early stage from 0 ~ 1.52 hours, and thereafter begins to decrease. It is known that CO₂-to-HCO₃⁻ conversion occurs during the reaction process, and the [Na⁺][HCO₃⁻] reaches a K_{sp} (1.30); so it is proposed that NaHCO₃ begin to precipitate at 1.52 hours. With the elongation of reaction duration, the Na⁺ concentration sharply decreases, which is owing to accelerated NaHCO₃ solid precipitates with more HCO₃⁻ formed during this process. Finally, when the Na⁺ concentration in the solution decreases to ~ 0.22 mol L⁻¹, the system stops precipitating NaHCO₃ (15.5 hours).

Based on the above data, we have plotted the change of $\text{Na}^+/\text{NH}_4^+$ concentration ratios with reaction duration. We have corrected all of the electrochemical data according to the reaction durations (Supplementary Fig. 69c, Figure 2e–g, Figure 4b and Figure 4d).

2) Following the reviewer’s suggestion, comparison experiments of using $2 \text{ mol L}^{-1} \text{ Na}_2\text{SO}_4$ electrolyte were conducted (Figure 2g). At current densities ranging from 100 to 500 mA cm^{-2} , the yield was 63.9, 122.8, 162.8, 206.1 and $254.0 \text{ mmol L}^{-1} \text{ h}^{-1} \text{ cm}^{-2}$, respectively. The performance is lower than that observed in the NaNO_3 electrolyte, which was 87.5, 156.3, 251.9, 313.6 and $374.4 \text{ mmol L}^{-1} \text{ h}^{-1} \text{ cm}^{-2}$, respectively. This result further confirmed that NO_3^- reduction (NO_3RR) indeed could accelerate the generation of NaHCO_3 in *operando*-electrified Solvay process.

Based on the above results, we have revised/added the following sentences and figures in the updated manuscript:

Figure 2g. comparison experiments for producing NaHCO_3 in $2 \text{ M Na}_2\text{SO}_4$ solution with CO_2 pumping and 2 M NaNO_3 solution with Ar pumping, respectively.

Supplementary Fig. 69. **a**, the schematic diagram of cyclic system for HCO_3^- accumulation. **b**, the simulated concentration of Na^+ , NH_4^+ , HCO_3^- and NO_3^- with reaction durations. **c**, the change of $\text{Na}^+/\text{NH}_4^+$ concentration ratios with reaction duration, which is an indication of $\text{NaHCO}_3/\text{NH}_4\text{HCO}_3$ ratio.

The calculation details are listed as follows: the initial solution was 50 mL of 2M NaNO_3 . Following the durability test, the applied current was assumed to be 2 A, and yield rate of HCO_3^- was $0.43 \text{ mol L}^{-1} \text{ h}^{-1}$. Thereby, the yield rate of NH_4^+ was calculated according to Faraday's law:

$$v_{\text{NH}_4^+} = \frac{I \cdot \eta \cdot 3600}{8 \cdot F \cdot V_0} \quad (46)$$

Where $v_{\text{NH}_4^+}$ means yield rate. I , η and F represent current, FE and faraday constant, respectively. V_0 is the volume of electrolyte.

Step 1: no precipitation

$$[\text{NO}_3^-]_t = [\text{NO}_3^-]_{t-\Delta t} - v_{\text{NH}_4^+} \cdot \Delta t \quad (47)$$

$$[Na^+]_t = [Na^+]_{t-\Delta t} \quad (48)$$

$$[NH_4^+]_t = [NH_4^+]_{t-\Delta t} + v_{NH_4^+} \cdot \Delta t \quad (49)$$

$$[HCO_3^-]_t = [HCO_3^-]_{t-\Delta t} + v_{HCO_3^-} \cdot \Delta t \quad (50)$$

Where $[NO_3^-]$, $[Na^+]$, $[NH_4^+]$ and $[HCO_3^-]$ are the ion concentrations in electrolyte, t and Δt are the reaction time and time step, respectively. The $v_{HCO_3^-}$ is yield rate of HCO_3^- .

Step 2: NaHCO₃ precipitation

The concentration of saturated NaHCO₃ solution is 1.14 mol L⁻¹ at ambient condition, thus, the $K_{sp,NaHCO_3}$ of NaHCO₃ was calculated to be 1.30. The amount of precipitation was set to be s .

$$([Na^+]_t - s) \cdot ([HCO_3^-]_t - s) \geq K_{sp,NaHCO_3} \quad (51)$$

The initial critical condition that has begun to precipitate has been calculated as:

$$s^2 - ([Na^+]_t + [HCO_3^-]_t)s + ([Na^+]_t \cdot [HCO_3^-]_t - K_{sp,NaHCO_3}) = 0 \quad (52)$$

$$s = \frac{([Na^+]_t + [HCO_3^-]_t) - \sqrt{([Na^+]_t + [HCO_3^-]_t)^2 - 4([Na^+]_t \cdot [HCO_3^-]_t - K_{sp,NaHCO_3})}}{2} \quad (53)$$

Thus, the concentrations of Na⁺ and HCO₃⁻ were obtained.

$$[HCO_3^-] = [HCO_3^-]_t - s \quad (54)$$

$$[Na^+] = [Na^+]_t - s \quad (55)$$

The concentrations of NH₄⁺ and NO₃⁻ changed as a constant rate, respectively.

It is seen from above figure that the ratio of Na⁺/NH₄⁺ concentration changes continuously with the elongation of reaction duration. The initial NaNO₃ electrolyte contains of NO₃⁻ (2 mol L⁻¹) and Na⁺ (2 mol L⁻¹). With the polarization at 2 A from 0 ~ 15 hours, the concentrations of NO₃⁻ decrease linearly from 2 mol L⁻¹ to 0.6 mol L⁻¹, while that of NH₄⁺ increases linearly from 0 mol L⁻¹ to 1.4 mol L⁻¹, which is in line with the NO₃⁻-to-NH₄⁺ conversion.

In great contrast, the concentration of Na^+ remains stable at 2 mol L^{-1} in the early stage from $0 \sim 1.52$ hours, and thereafter begins to decrease. It is known that CO_2 -to- HCO_3^- conversion occurs during the reaction process, and the $[\text{Na}^+]\cdot[\text{HCO}_3^-]$ reaches a K_{sp} (1.30); so it is proposed that NaHCO_3 begin to precipitate at 1.52 hours. With the elongation of reaction duration, the Na^+ concentration sharply decreases, which is owing to accelerated NaHCO_3 solid precipitates with more HCO_3^- formed during this process. Finally, when the Na^+ concentration in the solution decreases to $\sim 0.22 \text{ mol L}^{-1}$, the system stops precipitating NaHCO_3 (15.5 hours).

Based on the above data, we have plotted the change of $\text{Na}^+/\text{NH}_4^+$ concentration ratios with reaction duration. We have corrected all of the electrochemical data according to the reaction durations. (discussion following Supplementary Fig. 69)

Figure 2e. the corresponding NaHCO_3 productivities

Figure 2f. the NaHCO_3 productivities of GaOOH/rGO, GaOOH and rGO catalysts at different applied potentials.

Figure 4b. the NaHCO₃ productivities of GaOOH/rGO at different electrode areas.

Figure 4d. the NaHCO₃ productivities of GaOOH/rGO catalysts with the electrode area of 5 × 5 cm² at different currents.

Original Comment 4. *The Faradaic efficiency of the main product NH₃ during NO₃RR should be measured. Moreover, since various by-products can be generated during NO₃RR, the authors also need to test the Faradaic efficiency of N-containing by-products, such as NO₂⁻, N₂H₄, etc.*

Response: Following the reviewer's comment, we have conducted additional tests to determine the NO₃RR yield and Faradaic efficiency (FE) of GaOOH/rGO, GaOOH, and rGO catalysts (Supplementary Fig. 16–21).

At a current density of 500 mA cm⁻², GaOOH/rGO catalyst achieved a NH₃ yield rate of 1.29 mmol cm⁻² h⁻¹ and Faradaic efficiency of 55.3%, surpassing the comparison samples of GaOOH (0.66 mmol cm⁻² h⁻¹, 28.2%) and rGO (0.78 mmol

$\text{cm}^{-2} \text{h}^{-1}$, 33.5%). Other by-products of NO_2^- and N_2H_4 have also been measured. The NO_2^- yield of GaOOH/rGO catalyst was $1.49 \text{ mmol cm}^{-2} \text{ h}^{-1}$ (Faradaic efficiency of 16.0%), while GaOOH was $1.74 \text{ mmol cm}^{-2} \text{ h}^{-1}$ (Faradaic efficiency of 18.6%) and rGO was $1.69 \text{ mmol cm}^{-2} \text{ h}^{-1}$ (Faradaic efficiency of 18.2%), respectively. The N_2H_4 yield of GaOOH/rGO catalyst was $0.0142 \text{ mmol cm}^{-2} \text{ h}^{-1}$ (Faradaic efficiency of 1.06%), while the GaOOH was $0.0145 \text{ mmol cm}^{-2} \text{ h}^{-1}$ (Faradaic efficiency of 1.09%) and rGO was $0.0144 \text{ mmol cm}^{-2} \text{ h}^{-1}$ (Faradaic efficiency of 1.08%), respectively.

Based on the above results, we have added the following sentences and figures to the revised manuscript and supplementary information:

“Firstly, NO_3RR can convert NO_3^- into NH_3 as the dominant product with minor amounts of NO_2^- and N_2H_4 (Supplementary Fig. 16–21).” (Page 8, line 13–Page 8, line 14)

“The concentration of NH_3 byproduct was determined by indophenol blue method with modification.⁶ Initially, a portion of electrolyte was extracted from the electrolytic cell and subsequently diluted to the detection range. Next, 2 mL of the diluted electrolyte was mixed with NaOH solution (2 mL, 1 M) containing of 5 wt% salicylic acid and 5 wt% sodium citrate, followed by the addition of 1 mL of 0.05 M NaClO and 0.2 mL of 1.0 wt% sodium nitroferricyanide solution. After two hours under ambient condition, the absorption peak was measured by UV-vis spectrophotometer at the wavelength of 655 nm.

The concentration of NO_2^- byproduct was determined by UV-vis spectrophotometer.⁶ Firstly, a color reagent was prepared as follows: 4 g of p-aminobenzenesulfonamide, 0.2 g of N-(1-naphthyl) ethylenediamine dihydrochloride and 10 mL of phosphoric acid were added into 50 mL of deionized water and mixed thoroughly. Subsequently, 5 mL of the diluted electrolyte and 0.1 mL of the color reagent were mixed for 20 minutes under ambient condition. The absorption spectrum was measured by UV-vis spectrophotometer at the wavelength of 540 nm.

The concentration of N_2H_4 byproduct was determined by Watt and Chrisp method.⁷ The color reagent was prepared by 5.99 g of para-(dimethylamino)benzaldehyde, 30 mL of concentrated hydrochloric acid and 300 mL

of ethanol. Subsequently, 5 mL of the diluted electrolyte and 5 mL of color reagent were mixed for 10 minutes under ambient condition. The absorption spectrum was measured by UV-vis spectrophotometer at the wavelength of 457 nm.” (Supplementary Methods, Determination of nitrate reduction reaction (NO₃RR) product)

Supplementary Fig. 16. UV-vis calibration curve of NH₃. **a**, the UV-vis spectra of standard NH₄Cl solution with different concentrations. **b**, fitted calibration curve used for estimating NH₃ byproduct.

Supplementary Fig. 17. The NH₃ yield rate and Faradaic efficiency for GaOOH/rGO, GaOOH and rGO samples at a current density of 500 mA cm⁻². **a**, NH₃ yield rate. **b**, Faradaic efficiency of NH₃ yield. At a current density of 500 mA cm⁻², GaOOH/rGO achieved a NH₃ yield rate of 1.29 mmol cm⁻² h⁻¹ and Faradaic efficiency of 55.3%, outperforming the comparison samples of GaOOH (0.66 mmol cm⁻² h⁻¹, 28.2%) and rGO (0.78 mmol cm⁻² h⁻¹, 33.5%).

Supplementary Fig. 18. UV-vis calibration curve of NO_2^- . **a**, the UV-vis spectra of standard NaNO_2 solution with different concentrations. **b**, fitted calibration curve used for estimating NO_2^- byproduct.

Supplementary Fig. 19. The NO_2^- yield rate and Faradaic efficiency for GaOOH/rGO, GaOOH and rGO samples at a current density of 500 mA cm^{-2} . **a**, yield rate. **b**, Faradaic efficiency. The NO_2^- yield of GaOOH/rGO was $1.49 \text{ mmol cm}^{-2} \text{ h}^{-1}$ (Faradaic efficiency of 16.0%), while GaOOH was $1.74 \text{ mmol cm}^{-2} \text{ h}^{-1}$ (Faradaic efficiency of 18.6%) and rGO was $1.69 \text{ mmol cm}^{-2} \text{ h}^{-1}$ (Faradaic efficiency of 18.2%), respectively.

Supplementary Fig. 20. UV-vis calibration curve of N_2H_4 . **a**, the UV-vis spectra of standard N_2H_4 solution with different concentrations. **b**, fitted calibration curve used for estimating N_2H_4 byproduct.

Supplementary Fig. 21. The N_2H_4 yield rate and Faradaic efficiency for GaOOH/rGO, GaOOH and rGO samples at a current density of $500 mA cm^{-2}$. **a**, yield rate. **b**, Faradaic efficiency. The N_2H_4 yield of GaOOH/rGO was $0.0142 mmol cm^{-2} h^{-1}$ (Faradaic efficiency of 1.06%), while the GaOOH was $0.0145 mmol cm^{-2} h^{-1}$ (Faradaic efficiency of 1.09%) and rGO was $0.0144 mmol cm^{-2} h^{-1}$ (Faradaic efficiency of 1.08%), respectively.

Original Comment 5. *The stability test was conducted at a current of 2 A, which corresponds to a very small current density ($2 \text{ A} / 25 \text{ cm}^2 = 80 \text{ mA cm}^{-2}$), but the economic cost calculation is based a much higher current density (240 mA cm^{-2}), so how about the stability at this high current density?*

Response: Thanks for your useful comment. We have added the stability test at the current density of 240 mA cm^{-2} . As shown in Supplementary Fig. 34–35, the system has demonstrated stable performance with little potential alteration over 100 hours. The NaHCO_3 concentrations have been measured at approximately every 12 hours, which are 0.173 mol L^{-1} for 12 hours, 0.182 mol L^{-1} for 24 hours, 0.173 mol L^{-1} for 36 hours, 0.164 mol L^{-1} for 48 hours, 0.169 mol L^{-1} for 60 hours, 0.164 mol L^{-1} for 72 hours, 0.165 mol L^{-1} for 84 hours and 0.164 mol L^{-1} for 96 hours, respectively.

Further, we have examined the GaOOH/rGO electrode's performance before and after the 100-hour stability test (Supplementary Fig. 35). By operation for 100 hours, the productivity is $197.5 \text{ mmol L}^{-1} \text{ h}^{-1} \text{ cm}^{-2}$ at a current density of 240 mA cm^{-2} . After a long-term stability test, there was only 5% degradation. Therefore, our catalyst and system are stable for practical applications.

Based on the above results, we have added the following sentences and figures to the updated manuscript:

“Moreover, the GaOOH/rGO electrode exhibited durability exceeding for 100 hours (Supplementary Fig. 34–35).” (Page 9, line 10–Page 9, line 11)

Supplementary Fig. 34. **a**, the 100-hour durability test of GaOOH/rGO catalyst at 240 mA cm^{-2} , operating in a single-pass electrolyte mode. **b**, the accumulated NaHCO_3

concentration during durability test at a 12-hour interval. The system has demonstrated stable performance with little potential alteration over 100 hours. The NaHCO_3 concentrations have been measured at every 12 hours, which are 0.173 mol L^{-1} for 12 hours, 0.182 mol L^{-1} for 24 hours, 0.173 mol L^{-1} for 36 hours, 0.164 mol L^{-1} for 48 hours, 0.169 mol L^{-1} for 60 hours, 0.164 mol L^{-1} for 72 hours, 0.165 mol L^{-1} for 84 hours and 0.164 mol L^{-1} for 96 hours, respectively.

Supplementary Fig. 35. The productivity of NaHCO_3 for GaOOH/rGO electrode in 10-minute multiple-cycle tests before and after 100-hour durability test at 240 mA cm^{-2} .

Original Comment 6. *The calibration curve of urea measurement should be provided.*

Response: Please find the standard curves for urea calibration in the supplementary information as follows:

Supplementary Fig. 26. UV-vis calibration curve of urea. **a**, the UV-vis spectra of

standard urea solution with different concentrations. **b**, fitted calibration curve used for estimating urea byproduct.

Supplementary Fig. 27. The urea yield rate and Faradaic efficiency for GaOOH/rGO, GaOOH and rGO samples at a current density of 500 mA cm⁻². **a**, urea yield rate. **b**, Faradaic efficiency.

Original Comment 7. In the scheme showing the electrofixation of CO₂ and NO₃⁻ (Fig.2b), I suggest the same color to be used for the same element, which can make it easier understanding.

Response: Thanks for the kind reminding. We have revised the scheme as follows:

Figure 1b, the electrofixation of CO₂ and NO₃⁻ in flow cells.

Original Comment 8. How much is the conversion rate of CO₂? Is CO₂ possible be reduced to other compounds, like COOH?

Response: Thanks for the kind suggestion. Firstly, the conversion rate of CO₂ is 70%

under the conditions of 500 mA cm^{-2} & $10 \text{ mL min}^{-1} \text{ CO}_2$, which is calculated as follows:

$$\text{CO}_2 \text{ conversion rate} = \frac{C \times V}{r \times t / V_m}$$

Where C represents the production concentration in the electrolyte, V means the volume of electrolyte, r and t refer to gas flow rate and reaction time, respectively. And V_m is the molar volume of gas (22.4 L mol^{-1} under standard conditions). According to the above conditions, the values of C, V, r and t are $374.4 \text{ mmol L}^{-1}$, 50 mL, 10 mL min^{-1} and 1 hour, respectively. The final conversion rate of CO_2 is calculated to be 70%.

Secondly, we have also conducted nuclear magnetic resonance ($^1\text{H NMR}$) to test the electrolyte after reaction. The results showed no peak signals in the chemical shift range of 8.2–8.5, indicating no other byproduct (like COOH^-) generated from CO_2RR .

Based on the above results, we have added the following sentences and figures to the updated manuscript:

“where only trace gaseous H_2 and CO byproducts were detected (total Faradaic efficiencies $< 14.6\%$), with no other CO_2RR products (like HCOOH ; Supplementary Fig. 23–25).” (Page 8, line 15–Page 8, line 17)

Supplementary Fig. 25. The $^1\text{H NMR}$ spectrum of measured electrolytes for GaOOH/rGO, GaOOH and rGO at a current density of 500 mA cm^{-2} .

“The conversion rate of CO₂ (or CO₂ selectivity) is 70% under the conditions of 500 mA cm⁻² & 10 mL min⁻¹ CO₂, which is calculated as follows:

$$\text{CO}_2 \text{ conversion rate} = \frac{C \times V}{r \times t / V_m} \quad (56)$$

Where C represents the production concentration in the electrolyte, V means the volume of electrolyte, r and t refer to gas flow rate and reaction time, respectively. And V_m is the molar volume of gas (22.4 L mol⁻¹ under standard conditions). According to the above conditions, the values of C, V, r and t are 374.4 mmol L⁻¹, 50 mL, 10 mL min⁻¹ and 1 hour, respectively. The final conversion rate of CO₂ is calculated to be 70%.” (discussion following Supplementary Table 6)

Original Comment 9. *In Fig. 1a, in the history of NaHCO₃ synthesis, the authors show the first discovery of different synthesis methods. While for the modern electrified Solvay process, they only wrote “This work”, which may mislead readers to think this work is the first electrified Solvay process. Therefore, I suggest the authors to add the time or people of the first reported electrified Solvay process.*

Response: Thanks for the kind reminding. We have retrieved the literature and found the first report of the electrified Solvay process in 1985 by F. Hine et al *J. Electrochem. Soc.* 132, 2336 (1985). So we added this time point in Figure 1a as follows:

Figure 1a. the history of NaHCO₃ synthesis.

Original Comment 10. *In Supplementary Fig.13, CO is detected in all three samples, but in Supplementary Fig.15, the FE of CO is zero. Why?*

Response: Thanks for the kind reminding. The Faradaic efficiency of CO byproduct is less than 0.03%, which is unobvious in the original figure as comparison to H₂ (> 10%).

Now, we have replotted the data in Supplementary Fig. 23, containing only CO as follows:

Supplementary Fig. 23. The CO product determination by FID for GaOOH/rGO, GaOOH and rGO at a current density of 500 mA cm⁻². **a**, the detected curves of FID. **b**, the calculated Faradaic efficiency of CO.

Original Comment 11. *The authors wrote, “seldom urea has been detected in this work that indicates the fixation between CO₂ and NO₃⁻ instead of C–N coupling (Supplementary Fig. 12–16), with only trace gaseous H₂ and CO byproducts detected in the whole process (total Faradaic efficiencies < 10%)”. However, in Supplementary Fig. 15, it can be clearly seen that the Faradaic of H₂ exceeds 10% at some current density. Please modify this statement.*

Response: Thanks for the kind reminding. We have rechecked the data of Supplementary Fig. 12–16, and modified the statement as follows:

“where only a small amount of gaseous H₂ and CO byproducts were detected (total Faradaic efficiencies < 14.6%), with no other CO₂RR products (like HCOOH; Supplementary Fig. 23–25).” (Page 8, line 15–Page 8, line 17)

Original Comment 12. *Please carefully check the language. There are a lot of grammar errors and typos.*

Response: Thanks for the kind reminding. We have rechecked the whole manuscript and corrected typos and errors as follows:

“In this work, we leverage the ‘octet rule’, a fundamental principle in elemental chemistry, to address the above challenging problem.” (Page 5, line 1–Page 5, line 2)

“Other than hydrogen chemistry, such as HER, a vast range of elements in the periodic table adhere to the octet rule, allowing them to accommodate up to eight electrons in valence shells.” (Page 5, line 3–Page 5, line 5)

“This characteristic could facilitate maximally eight-electron chemical reactions.” (Page 5, line 5–Page 5, line 6)

“Motivated by the hypothesis, we developed an operando-electrified system of delivering two environmental pollutants (CO_2 and NO_3^-) into an electrochemical cell to produce NaHCO_3 (Fig. 1b–c) despite its productivity still unable to rival Solvay process, owing to the stumbling scaling relationship (the relationship between productivity and production scale) inside the updated system” (Page 5, line 11–Page 5, line 15)

“Here, we have calculated the $\ast\text{CO}$ adsorption energy of liquid metal gallium (Ga), and found it situated at a distance far from the volcano apex ($E_{\ast\text{CO}} = -1.10$ eV and corresponding to the limiting potential of -1.09 V)” (Page 6, line 17–Page 6, line 19)

“therefore, it would be an inert candidate for CO_2RR . Nevertheless, as will be discussed later, Ga active sites have still displayed a physical adsorption to CO_2 , which can serve to enrich CO_2 on the electrode surface and facilitate the chemical CO_2 -to- HCO_3^- conversion.” (Page 6, line 19–Page 6, line 22)

“On the other hand, CO_2 in gas chamber enters into cathode chamber *via* the gas diffusion electrodes” (Page 7, line 20–Page 7, line 21)

“The overall productivity shows a nearly linear correlation relationship to electrode areas (productivity = $0.12 \ast \text{area} + 0.15$, Fig. 4b), highlighting the advantages of our module cell in contributing to maximizing the use of electrode surfaces.” (Page 12, line 14–Page 12, line 16)

“This value initializes at $1.04 \text{ mol L}^{-1} \text{ h}^{-1}$ at the small flow rate of 1.7 mL min^{-1} owing to insufficient feedstock supply” (Page 12, line 18–Page 12, line 19)

Response to Reviewer #1:

Original Comment: *The authors have made substantial enhancements to the manuscript, particularly in emphasizing process innovation and validating CO₂ adsorption mechanisms through new in situ characterizations. These revisions significantly strengthen the work's industrial relevance and mechanistic credibility, aligning perfectly with previous review suggestions. Authors can consider the following minor remaining issues:*

Original Comment 1: *Clarify Current Density Units. Standardize current density in Fig. 4d (currently uses “A” without area specification).*

Response: Thanks for your kind reminding. We have standardized the unit of applied current density in Figure 4 as follows:

Figure 4d, the NaHCO₃ productivities of GaOOH/rGO catalysts with the electrode area of 5 × 5 cm² at different current densities.

Original Comment 2: *Theoretical Justification. Cite the octet rule principle in the Introduction.*

Response: The relevant literature has been cited as ref. 20 as follows:

“In this work, we leverage the ‘octet rule’,²⁰ a fundamental principle in elemental chemistry, to address the above challenging problem.” (Page 5, line 1–Page 5, line 2)

Ref. 20. Bent, H. A. An appraisal of valence-bond structures and hybridization in compounds of the first-row elements. *Chem. Rev.* 61, 275–311 (1961).

Original Comment 3: *Economic Assumptions. Briefly note the \$0.03/kWh electricity cost rationale in the main text (currently only in SI).*

Response: Thanks for your kind reminding. We have added relevant description as follows:

“Further economic evaluations demonstrate the production cost of the proposed modular cell. Based on renewable electricity priced at \$0.03 kWh⁻¹,⁴⁴ the operational cost of the proposed *operando* process is much lower than that of traditional/improved Solvay processes (\$1829.7 vs. \$4106.1 ton⁻¹ day⁻¹), representing a more economically viable option (Fig. 4h; Supplementary Fig. 74).” (Page 13, line 17–Page 13, line 20)

Ref. 44. Zhang, L. C., et al. High-efficiency ammonia electrosynthesis from nitrate on ruthenium-induced trivalent cobalt sites. *Energy Environ. Sci.* 18, 5622–5631 (2025).

Response to Reviewer #5:

Original Comment: *In the response to my original Comment 1, the author said that the potential for GaOOH/rGO to achieve 500 mA cm⁻² is -2.00 V (which is also shown in Figure 2d). However, in Figure 2e and 2f, the productivity of GaOOH/rGO at 500 mA cm⁻² (Figure 2e) is much higher than that at -2.00 V (Figure 2f). Please provide explanations for this large discrepancy.*

Response: Thanks for your kind reminding. Based on Reviewer's comment, we have repeated the experiments for Figures 2d,2e,2f, and got similar results. Thereby, we have retrieved the literature, and attributed the discrepancy to different electrochemical testing techniques, *i.e.*, chronoamperometry (Figures 2d, 2e) and chronopotentiometry (Figure 2f). We give more detailed explanations as follows:

- 1) Generally, chronopotentiometry uses fixed current as controlled variable and potential as response variable. This is different from chronoamperometry with fixed applied potential as controlled variable and current as response variable. Ideally, the results obtained by these two techniques should be the same because of they reflecting the performance of one electrocatalytic system.
- 2) Nevertheless, under actual electrochemical condition, applying these two techniques can cause different alterations in system parameters, leading to discrepant test results. The relevant system parameters mainly include electrode surface states (active sites, deposition of by-products), mass transfer (reactant supply under high current), and ohmic drop.
- 3) As a result, in our electrochemical system, the amount of reaction transferred coulombs is used as the benchmark to evaluate reaction performance (Nat Commun 2025, 16, 5742; Nat Commun 2024,15, 678). As shown in Supplementary Fig. 15d, the reaction coulombs of GaOOH/rGO, GaOOH and rGO under chronoamperometry testing at 2 V (*vs.* RHE) are 215.7 C, 183.2 C and 175.5 C, which correspond to productivities of 293.0, 228.3 and 218.8 mmol L⁻¹ h⁻¹ cm⁻², respectively (Figure 2f). On the other hand, in chronopotentiometry tests (Figure 2d-e; Supplementary Fig. 12-13), the coulombs transferred at 500 mA cm⁻² are

constant at 300 C, leading to the productivities of 374.4, 343.1 and 355.8 mmol L⁻¹ h⁻¹ cm⁻².

- 4) Further, we have surveyed literature and found also many previous studies with discrepant results in chronoamperometry and chronopotentiometry results, including nitrate reduction reaction (Nat Commun 2025, 16, 5742, Figure R1), oxygen reduction reaction (Nat Commun 2024, 15, 678), Figure R2), organic oxidation reaction (Nat Commun 2025, 16, 286), Figure R3; ACS Appl. Mater. Interfaces 2025, 17, 6, 9391, Figure R4), and carbon dioxide reduction reaction (ACS Appl. Mater. Interfaces 2024, 16, 14, 17371, Figure R5). All of the productivities have been evaluated by the amount of reaction transferred coulombs as the benchmark.
- 5) To avoid confusion, we have clearly mentioned chronopotentiometry or chronoamperometry tests in corresponding Figure legends, and made the following revisions:

Supplementary Fig. 15. The **chronoamperometry** tests at different potentials (vs. RHE) in 2 M NaNO₃ aqueous solution with CO₂ pumping for **a**, GaOOH/rGO catalyst, **b**, GaOOH catalyst and **c**, rGO catalyst, respectively. **d**, reaction coulombs of GaOOH/rGO, GaOOH and rGO under **chronoamperometry** tests at 2V vs. RHE

Figure 2d, the chronopotentiometry tests of GaOOH/rGO.

Figure 2e, the corresponding NaHCO₃ productivities in chronopotentiometry tests of GaOOH/rGO.

Figure 2f, the chronoamperometry tests and corresponding productivities of GaOOH/rGO, GaOOH and rGO catalysts.

Supplementary Fig. 12. **a**, The **chronopotentiometry** tests of GaOOH catalyst at different current densities in 2M NaNO₃ aqueous solution with CO₂ pumping. **b**, The **chronopotentiometry** tests of rGO catalyst at different current densities in 2 M NaNO₃ aqueous solution with CO₂ pumping. **c**, The average potentials of GaOOH/rGO, GaOOH and rGO catalysts at different current densities in **chronopotentiometry** tests.

Supplementary Fig. 13. The NaHCO₃ productivity in **chronopotentiometry** tests. **a**, GaOOH catalyst in 2 M NaNO₃ aqueous solution with CO₂ pumping. **b**, rGO catalyst in 2 M NaNO₃ aqueous solution with CO₂ pumping.

Figure R1. **a**, Chronopotentiometry curve of hcp-Ru₁Co in an H-type continuous-flow cell at 20 mA cm⁻² with corresponding Y_{NH₃} and FE_{NH₃}. **b**, j_{NH₃} of hcp-Ru₁Co, hcp/fcc-Ru₁Co, and fcc-Ru₁Co at different potentials in 0.5 M Na₂SO₄ and 0.1 M NaOH with 200 ppm NO₃⁻-N (Nat Commun 16, 5742 (2025)).

Note. The Faradaic efficiency/yield rates are 90%/1.4 mg h⁻¹ cm⁻² calculated from chronopotentiometry, while 95%/0.9 mg h⁻¹ cm⁻² calculated from chronoamperometry.

Figure R2. **a**, The chronoamperometry measurements at varied applied voltages of CoPc-S-COF. **b**, Chronopotentiometry curve at a current density of 125 mA cm⁻² and the corresponding FE_{H₂O₂} in the flow cell for CoPc-S-COF (Nat Commun 15, 678 (2024)).

Note. The Faradaic efficiency is 94% calculated from chronopotentiometry, while 97% calculated from chronoamperometry.

Figure R3. **a**, Comparison of the applied potentials to reach different current densities for Ni₃FeN-Ru_{buried}, Ni₃FeN-Ru_{surface}, and Ni₃FeN. **b**, Chronoamperometry curve of the stability test (Nat Commun 16, 286 (2025)).

Note. The Faradaic efficiency is 50% calculated from chronopotentiometry, while 73% calculated from chronoamperometry.

Figure R4. **a**, Chronopotentiometric study at two different potentials (25 mA for HER and 25 mA for OER) with 3% Pt/NiW. **b**, CA-HER study at different potentials of -0.15 , -0.25 , -0.35 , and -0.45 V (vs RHE) (ACS Appl. Mater. Interfaces 2025, 17, 6, 9391–9406).

Note. The current/potential are -25 mA/ -0.5 V from chronopotentiometry, while -25 mA/ -0.35 V from chronoamperometry.

Figure R5. **a**, Chronopotentiometric study at two different potentials (25 mA for HER and 25 mA for OER) with 3% Pt/NiW. **b**, CA-HER study at different potentials of -0.15 , -0.25 , -0.35 , and -0.45 V (vs RHE) (ACS Appl. Mater. Interfaces 2024, 16, 14, 17371–17376).

Note. The Faradaic efficiency is 48% calculated from chronopotentiometry, while 25% calculated from chronoamperometry.

Comments

In this work, the authors present an interesting method to produce NaHCO_3 with the assistance of electrocatalytic nitrate reduction. The productivity of HCO_3^- is much higher than the traditional Solvay process. However, I feel there are some issues that the authors need to carefully address. My detailed comments are as follows.

1. The authors claimed that there is a “strong synergistic effect between GaOOH and rGO ”. However, the electrocatalytic performance of GaOOH , rGO , and GaOOH/rGO do not show much difference (Fig.2e, Supplementary Fig.9-10). It seems that the synergistic effect between GaOOH and rGO is weak.
2. How about the purity of the generated NaHCO_3 ? The product purity is closely related to the purification cost, which should also be taken into account when calculating the economic costs. Btw, I suggest the authors to provide the details for the product collection and purification.
3. During the CO_2 -assisted electrocatalytic NO_3^- reduction, NH_4^+ will be produced in the aqueous solution. This means that, NaHCO_3 and NH_4HCO_3 are produced simultaneously. For the calculation of HCO_3^- productivity, the authors measure the total concentration of HCO_3^- in the solution, which comes from both NaHCO_3 and NH_4HCO_3 , but only NaHCO_3 is the target product. So I think calculating NaHCO_3 productivity is more reasonable. Similarly, the authors should compare the productivity of NaHCO_3 in CO_2 -purged NaNO_3 and NaSO_4 solution, to prove that NO_3^- reduction indeed accelerate the generation of NaHCO_3 .
4. The Faraday efficiency of the main product NH_3 during NO_3RR should be measured. Moreover, since various by-products can be generated during NO_3RR , the authors also need to test the Faraday efficiency of N-containing by-products, such as NO_2^- , N_2H_4 , etc.
5. The stability test was conducted at a current of 2 A, which corresponds to a very small current density ($2 \text{ A} / 25 \text{ cm}^2 = 80 \text{ mA cm}^{-2}$), but the economic cost calculation is based a much higher current density (240 mA cm^{-2}), so how about the stability at this high current density?
6. The calibration curve of urea measurement should be provided.
7. In the scheme showing the electrofixiation of CO_2 and NO_3^- (Fig.2b), I suggest the same color to be used for the same element, which can make it easier understanding.
8. How much is the conversion rate of CO_2 ? Is CO_2 possible be reduced to other compounds, like COOH^- ?
9. In Fig.1a, in the history of NaHCO_3 synthesis, the authors show the first discovery of different synthesis methods. While for the modern electrified Solvay process, they only wrote “This work”, which may mislead readers to think this work is the first electrified Solvay process. Therefore, I suggest the authors to add the time or people of the first reported electrified Solvay process.
10. In Supplementary Fig.13, CO is detected in all three samples, but in Supplementary Fig.15, the FE of CO is zero. Why?

11. The authors wrote, “seldom urea has been detected in this work that indicates the fixation between CO_2 and NO_3^- instead of C-N coupling (Supplementary Fig. 12-16), with only trace gaseous H_2 and CO byproducts detected in the whole process (total Faradaic efficiencies $< 10\%$)”. However, in Supplementary Fig. 15, it can be clearly seen that the Faraday of H_2 exceeds 10% at some current density. Please modify this statement.

12. Please carefully check the language. There are a lot of grammar errors and typos.

1. In the response to my original Comment 1, the author said that the potential for GaOOH/rGO to achieve 500 mA cm^{-2} is -2.00 V (which is also shown in Figure 2d). However, in Figure 2e and 2f, the productivity of GaOOH/rGO at 500 mA cm^{-2} (Figure 2e) is much higher than that at -2.00 V (Figure 2f). Please provide explanations for this large discrepancy.